



# Climatic extremes and their social impact in 17th-century Transylvania. A climate-historical reconstruction in the context of the Little Ice Age

Ovidiu Razvan Gaceu[1], Tudor Caciora[1]*, Mihai Dudaș[2], Ștefan Baias[1], Marius Stupariu[1], Maria Maxim[1], Cătălina Mărculeț[3]

[1]Department of Geography, Tourism and Territorial Planning, Faculty of Geography, Tourism and Sport, University of Oradea, Oradea 410087, Romania
[2]Jimbolia Technology Highschool, Jimbolia 305400, Romania
[3]Institute of Geography of the Romanian Academy of Science, Bucharest 023993, Romania

*Correspondence to*: Tudor Caciora (tudor.caciora@yahoo.com)

**Abstract.** The 17[th] century represented one of the most difficult climatic and social periods in the recent history of Europe, being marked by the transition to the Maunder Minimum, the peak of the Little Ice Age; marked by low thermal values, long

and severe winters, rainy and cool summers, numerous dangerous climatic phenomena, a reduced number of normal years and seasons, famines, epidemics, etc. Taking these aspects into account, the present study aims to reconstruct, based on written historical sources, the climatic variability in the Principality of Transylvania during the 17[th] century. The emphasis is placed on the climatic phenomena generated by temperature and precipitation, as well as on the calamities associated with climate and their impact on society. The analysis identified a high prevalence of events associated with cold weather, 54

winters being mentioned as particularly cold in this century, of which 36 occurred during the Maunder Minimum (1645-1715). This interval was also characterized by numerous episodes of excessive precipitation and a clear imbalance between events associated with excessive humidity and drought. Variable climatic conditions caused poor harvests, famine, high prices and favoured the emergence of severe epidemics and social crises. Correlating historical sources with proxy data from the natural archive confirms the exceptional character of this century and validates the reconstruction based on historical

information. They also indicate that natural factors such as solar activity and general atmospheric circulation generated the colder and wetter climate in 17th-century Transylvania, especially during the winters of the Maunder Minimum.

**Keywords:** historical climatology, climate reconstruction, historical information, climate variability, Little Ice Age, Maunder Minimum




**1 Introduction**

The 17th century, in Europe, is marked by the Little Ice Age (LIA), a global cooling event that spanned a period of five to six centuries (from 1250-1350 to 1860) (Teodoreanu, 2017; Le Roy Ladurie, 1983; Le Roy Ladurie, 2004). During the LIA, winters were characterized by lower thermal values than today, while summers seem to have remained warm (Perșoiu and Perșoiu, 2018; Diodato et al., 2014). The variation in precipitation followed a more complex pattern than in the

case of the thermal regime. Most of the available data (from different proxy sources) suggest a transition from a Medieval Warm Period (MWP) (which occurred approximately between 850-900 to 1250-1350), relatively wet, to the LIA, characterized by drier conditions (Feurdean et al., 2015). In addition, analysis of macrofossils preserved in cave ice (Feurdean et al., 2011) suggests that although the LIA was generally a dry period, it was interrupted by episodes of intense precipitation. Some studies (Martín-Puertas et al., 2011) indicate that, in Europe, the precipitation regime in the LIA would

have been influenced in the long term by the North Atlantic Oscillation (NAO). In this regard, Perșoiu et al. (2017) mention that in the first part of the Holocene, the negative phase of the NAO (NAO-) would have acted more strongly on Central and South-Eastern Europe, while the second half of the Holocene is, for the most part, under the sign of the positive phase (NAO+), with a deficient precipitation regime.

The second half of this century is characterized by pronounced negative anomalies of winter temperature values, as

well as strong positive deviations of precipitation (Riedwyl et al., 2008a; Luterbacher et al., 2004; Shindell et al., 2001; Mann, 2002). During this period, the Maunder Minimum (MM), the coldest interval of the LIA period between approximately 1645 and 1715, is individualized. The MM, according to some authors, would have been characterized by a value of the average temperature in Europe approximately 2°C lower than today (Diodato & Bellocchi, 2012). Negative anomalies were manifested especially during the winter, while summer was characterized by slight positive anomalies, as

indicated by Riedwyl et al. (2008b) and Yamaguchi et al. (2010). The same author identified the last decade of the 17th century as the coldest in at least 500 years in Europe. The causes that triggered the MM remain debatable, with most authors indicating that reduced solar activity (during this period, several years without any sunspots recorded) was the main cause of the cold winters of this period (Shindell et al., 2001; Lockwood et al., 2010), followed by intense volcanic activity (Yoshimori et al., 2005; Sigl et al., 2015; Stoffel et al., 2021). Low solar activity would have weakened westerly winds and

favored the movement of cold air masses from the Arctic region to temperate latitudes, which led to colder winters and an increase in the frequency of extreme weather episodes in Europe and North America, especially in winter (Ineson et al., 2011). Recent studies on the solar influence on the jet stream blocking phenomenon (Barriopedro et al., 2008) are consistent with those above.

To reconstruct the climate of this century, both sources from the natural archive (tree rings, sporopollenic analyses,

cores, etc.) and data from the social archive (written documents, oral information, paintings, etc.) are considered (Baias et al., 2020; White et al., 2022). The latter include direct records of the weather, as well as indirect ones, such as harvest records, especially those relating to grain production and wine quality. Additional information about the social consequences





(increased prices, famine, decreased trade volume) can also be extracted from historical sources. Unfortunately, in chronicles
and annals, the information about meteorological conditions are sporadic and unsystematic, however, these references have
the advantage that extreme meteorological phenomena (e.g., abnormally hot or cold summers and winters, torrential rains,
long droughts, floods, etc.) are, in most cases, described precisely (Teodoreanu, 2014; Stangl & Foelsche, 2020). Information
from the society's archives, even if highly contested due to subjectivity and the high degree of difficulty in terms of
quantification, has the advantage of providing direct information, with a temporal and spatial resolution superior to proxies
and clarity in terms of their significance (Zheng et al., 2014; Chen et al., 2020; Carey, 2012; Stangl & Foelsche, 2020).

In western and central Europe, many studies that consider the use of written sources to reconstruct the climate in the
historical past attest to climatic phenomena and processes associated with a specific LIA and MM weather (Luterbacher et
al. 2000, Luterbacher et al. 2001, Bullón, 2020). At the same time, the best-known image associated with the LIA and MM is
presented by the Thames River, which frequently froze in the winter during the 17th and 18th centuries, allowing the
organization of fairs on the river, the original work belonging to the painter Abraham Hondius. Unlike in Western and
Central Europe, information about the manifestation of the LIA in Romania, in principle, and in Transylvania, in particular,
is very little known. Only a few studies have considered the reconstruction of the evolution of the climate based on
information from historical documents for this territory. Of particular note are the studies by Teodoreanu (2013, 2017),
Stangl & Foelsche (2020), Dudaș (1999, 2006, 2024), Dudaș and Urdea (2021), Mărculeț and Ștef (2005) and Gaceu et al.
(2025). Each of these authors makes an important contribution to better knowledge, but not sufficient, which is why the field
of study specific to historical climatology is in continuous development, through the discovery of new information and the
implementation of new techniques for quantifying and interpreting data.

Based on the above, the present study aims to reconstruct the climate of 17th-century Transylvania, based on
information from the society's archives, corroborated with established natural proxy databases. The hypothesis from which it
was started is that, in agreement with western and central Europe, the territory of Transylvania must have been characterized
by a similar climate, determining the LIA and MM, which could have generated social, economic and even political
imbalances in the principality. This is all the more interesting as studies indicate that the climate during the MM period, and
more broadly, during the LIA period, both at global and regional levels, was neither uniform nor constant [Yang and Jiang,
2017; Cronin et al., 2010; Neukom et al., 2019; Bertler et al., 2011]. Thus, the climate variations in this geographical region
were not identical to those in other European regions, but had distinct regional features, and by using local historical sources,
precisely these particularities can be highlighted. Although numerous studies aim to reconstruct the climate in Central and
Eastern Europe (especially the thermal and precipitation regimes) based on proxy data, no centralised studies aim to
characterize the climate and its implications in the 17th century in Transylvania based on society's archives. The use of these
documentary databases not only provides a significant contribution to understanding climate variability in the historical past,
when meteorological observations were not available, but also allows a detailed perspective on how climate phenomena
were perceived, felt and managed by individuals and communities. By highlighting human reactions to extreme climate



conditions, this complementary approach to proxy-based climatology brings to the fore the subjective and experiential dimension of climate, facilitating a deeper understanding of social adaptability in the face of climate-generated hardships.

## 2. Study area

### 2.1. Historical context

In the 17th century, Europe was going through a period of profound political, economic and social changes. The Thirty Years' War (1618-1648) devastated much of central Europe, with consequences for the balance of power between the great states. In Eastern Europe, the Ottoman Empire maintained its influence and even control over the Balkans and the Romanian Principalities, the latter under Ottoman suzerainty. Transylvania was not part, as it is today, of a unitary state
(Figure 1), being an autonomous principality under Ottoman suzerainty, but with greater internal freedom compared to Wallachia and Moldavia; starting with 1685, Transylvania was integrated into the structures of the Habsburg Empire, with the status of an autonomous principality (Giurescu, 1994). The 17th century was also marked by numerous wars between the great powers, especially between the Ottoman Empire and the Poles, Russians and Austrians, conflicts generally fought on the territories of the three Romanian Principalities, with heavy consequences for the inhabitants. At the same time, the
tributary obligations towards the Sublime Porte and the power games between the great neighbouring empires made the political and economic situation in the principalities difficult (Czamańska, 2023).

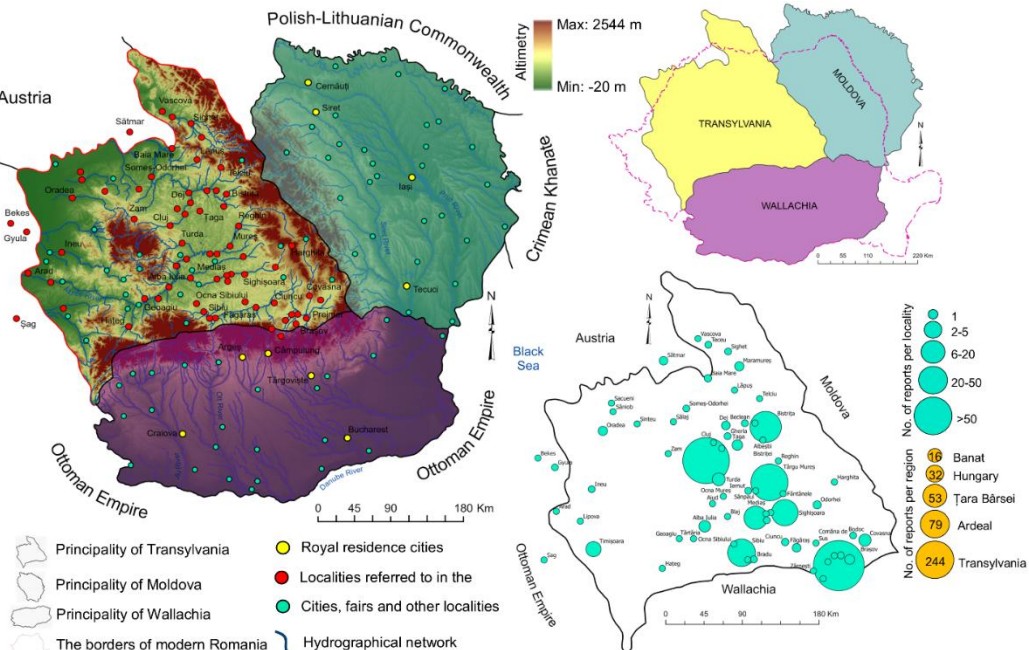

**Figure 1: The territorial expansion of the Romanian Principalities in the 17th century, the main localities of the time and the points of collection of information used in the present reconstruction**





In addition to all the aspects mentioned above, the population of the Romanian Countries, and Europe in general, also had to face the vagaries of the climate, which had a considerable impact on daily life and economic stability. Transylvania's privileged status, compared to the other Romanian Principalities, is also reflected in the quantity and quality of information sources, including those relating to climatic phenomena (Figure 1). In this context, a considerably larger volume of data from Transylvania is noted, which improves the scientific approach and offers a much more comprehensive overview of the climate in the 17th century in this geographical territory.

### 2.2. Climatic characterization of Romania

To better understand the information from historical documents regarding the climate of the 17th century in Transylvania, it is useful to correlate it with the characteristics of the current climate of Romania, to have a clearer picture of the intensity and impact of the described phenomena. The geographical location of Romania, on the globe and in Europe, along the 45º parallel of northern latitude, solar radiation and the circulation of air masses, determine a temperate continental transitional climate, with four seasons and with Baltic influences in the north, oceanic in the west, Mediterranean in the southwest, continental in the north and east and Pontic on the Black Sea coast. At the same time, the varied relief (with all three major steps being present) (Figure 1) generates different climatic conditions, both horizontally and vertically. Thus, the Transylvanian Depression was significantly influenced by the Carpathian Mountain arc, which surrounded it and acted as a barrier to the Atlantic, Baltic, Mediterranean, Pontic, and continental East European air masses.

Average temperatures vary depending on the region and altitude, from approximately 10-11°C in the Danube meadow and the plain, to 6-8°C in the foothills and less than 6°C at the base of the mountains, to -2°C on the highest ridges, the values decreasing from south to north (Clima României, 2008; Bogdan & Costea, 2013). In the Transylvanian Depression, the multiannual average values of the air temperature vary from 6°C at the contact with the mountains and up to 8-9°C in most of the depression, the highest values being on the hills affected by the fohn movements, namely the eastern slope of the Apuseni and on the lowest hills in the center. The absolute extreme values in Romania reached 44.5°C at Ion Sion, in 1951, and -38.5°C at Bod, in 1942 (Clima Romaniaiei, 2008).

Romania's location concerning the main pressure centres and the relief characteristics determine great differences in the distribution of atmospheric precipitation. At national level, the multiannual average amounts reach 600-700 mm/year, with minimum values of under 400 mm/year in the Danube Delta (267 mm/year at C.A. Rosetti) and Dobrogea and over 1600 mm/year at Stana de Vale (Western Carpathians) (Gaceu, 2012). Compared to these multiannual values, the annual ones varied significantly, being reduced by half in dry years (reaching 137.6 mm in 2000 at Sulina) or increased two-three times in rainy years (2370 mm at Stâna de Vale in 1980; 530.6 mm in 24 hours in 1924 in the Danube Delta) (Clima României, 2008; Teodoreanu, 2011; Gaceu 2012), with damaging consequences in agriculture, industry and in people's lives in general. In the Transylvanian Depression, the average annual precipitation amounts are between 600 and 800 mm, lower in the west of the depression and higher in the east.



### 3. Materials and methods

*3.1. Extracting climate information from documentary sources*

The databases used in the present study were both direct (primary) and indirect (secondary and tertiary) (Figure 2). The primary ones consider consulting original documents, written in the 17[th] century directly by the authors: chronicles, diaries, official and monastic registers, religious books, secular books, diaries of foreign travellers in the Romanian Countries, calendars and archives (Figure 2). Many of these documents are kept in Romanian and Hungarian archives, either 155 in original form or republished, representing an essential source of factual data for the detailed analysis of climate in the historical past. These sources are the most difficult to read and interpret because they are written predominantly in the old forms of the Hungarian, German (in the Saxon dialect), Slavic, Turkish or Latin languages, often with Cyrillic characters. Other disadvantages of these databases are that climate information is quite rare, and it is necessary to go through many pages to identify a relatively small volume of climate information.

Secondary sources are mainly old writings that directly document meteorological phenomena or indirectly provide valuable information about historical climatic events. Tertiary sources refer to synthesis works that provide an overview of historical events, with details on climatic aspects. Secondary and tertiary sources do not bring new information to the literature, but base their content on information provided by primary sources, which they analyze, interpret and integrate to provide a contextualized perspective on historical events (Figure 2).

Among the works consulted, worthy of mention, because they constitute true databases on climatic phenomena recorded over time in Transylvania, we must mention: Teodoreanu (2017), Topor (1964), Pap (1822), Hanusz (1890), Millhoffer (1898), Réthly (1962, 1970, 1998), Rácz (2001), Dudaş (1999, 2006), Stangl and Foelsche (2022). These authors used as primary databases mainly memoirs, diaries, chronicles, letters of personalities who lived in the 17[th] century, works of historians, naturalists and meteorologists published during the 19[th] and 20[th] centuries and notes made by priests on old 170 religious books from Transylvania. In addition to the works already mentioned, the present study is based on an extensive bibliographic corpus, references that will be mentioned in detail within the manuscript.

In the historical writings consulted, approximately 937 different testimonies were identified regarding the climate of Transylvania in the 17[th] century, recorded within 68 localities of the principality (Figure 1). The number of testimonies is unevenly distributed across the Transylvanian territory, depending on the literacy level of the local population. Thus, most 175 mentions of the climate are located in the large cities controlled by Hungarians and Saxons (Brasov, Sibiu, Cluj, Sighisoara, Medias, Bistrita, etc.). The notes most frequently come from representatives of these communities, who were, most of the time, literate people, with writing concerns. The majority of Romanians have the status of a tolerated nation, without access to schools, public administration, or politics, and the Romanian language and Orthodox faith are not recognised; this has caused the literacy rate to be low among Romanians. In some situations, the phenomena are not localised at the level of a 180 single city, fortress or village, but indicate that over an extended area (the whole of Transylvania, Ardeal, Tara Barsei, Banat,



etc.), the same climatic phenomenon was individualised. In this sense, no less than 244 testimonies concern the whole of Transylvania, 79 refer to Ardeal, and 53 to Tara Barsei (smaller territories within Transylvania) (Figure 1).

### *3.2. Codification and systematisation of historical data*

To reduce the uncertainties and subjectivity inherent in historical sources, a standardized methodology for coding and quantifying climate information was implemented in the work, in accordance with good practices established in the specialized literature by various authors (Butzer and Pfister, 1987; Brázdil et al., 2005; Rohr et al., 2018; Degroot et al., 2021).

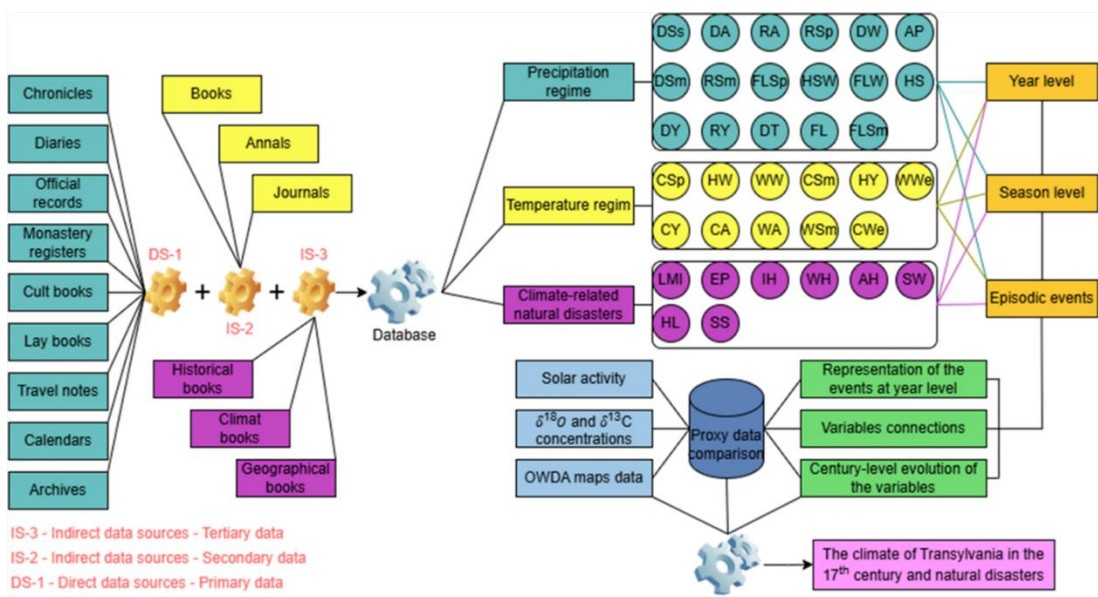

CY - cold year,  HY - hot year, CSp - cold spring, HW - harsh winter, WW - warm winter, CSm - cold summer, WSm - warm summer, CA - cold autumn, WA - warm autumn, DY - dry year, RY - rainy year, FL - floods, DT - droughts, FLSp - spring floods, DSs - dry spring, RSp - rainy spring, DSm - dry summer, FLSm - summer floods, RSm - rainy summer, DA - dry autumn, RA - rainy autumn, DW - dry winter, FLW - winter floods, HSW - Heavy snowfall in winter , LMI - Locust/mouse invasions, EP - epidemics, IH - inflation and hunger, WH - weak harvests, AH - abundant harvests, CWe - cold weather, WWe - warm weather, AP - abundant precipitation, HS - heavy snow, SW - strong winds, SS - strong storms, ND - no data

**Figure 2: Workflow and bibliographic sources used in the reconstruction of climatic phenomena from the 17ᵗʰ century in Transylvania**

A shortcoming of the present methodology is that the documents used as a data source were not written for climatological purposes; therefore, the presentation of climatic phenomena is not systematic. Thus, the information necessary to reconstruct the climate had to be extracted from the context based on testimonies relating to phenological

phases, the impact on agriculture (compromised or delayed harvests), the effects on transport and trade (impassable roads, freezing or drying up of rivers), as well as the social consequences of these phenomena (famines, epidemics, migrations, price increases), all of which provide indirect indications about the state of the weather and climatic conditions of the century. Thus, testimonies such as "*the wheat was harvested in June*" or "*the grapes were picked in August*" indicate warm weather, and statements such as "*many rivers and lakes froze to the bottom*" suggest a particularly cold winter in those

200     months. Regarding the precipitation regime, statements such as "*the grass caught fire in the fields and the forests burned*" or



"*the grapes and the sponges on the trees dried up*" suggest severe droughts, while "*the wine was plentiful, but sour*" indicates rainy weather. In addition to the information that must be extracted from the context, other writings indicate the exact type of phenomenon and its territorial and temporal extension, leaving no room for interpretation.

Thus, the climatic phenomena, although very diversely presented by those who observed (experienced) them, were simplified and attributed to synthetic categories, based on their origin and way of manifestation. This way, the database obtained is synthetic, much easier to interpret, and free from potential errors. At the same time, the narrative convergence of several independent sources was used as an essential criterion for internal validation. When the sources offered contradictory accounts, a territorial analysis of their location was used to highlight regional climatic variations or topoclimatic influences. Irreconcilable or isolated information was eliminated from the analysis (Brázdil et al., 2005; Pfister, 1996; Rohr et al., 2018). Based on the aforementioned bibliographic sources, the data were classified, depending on the climatic component, into three large categories: thermal variability, pluviometric variability, and climate-related natural disasters.

The thermal variability category includes references to cold or warm years and seasons and to cold or heat waves. Thus, 11 variables associated with this category were identified in the paper. They could not be classified into different items, given that they refer to phenomena limited to certain seasons or events of limited duration. The categories range from mentions of cold (CY) and warm (HY) years, to references to seasons, such as cold (HW) or warm (WW) winters, cold springs (CSp) and cold autumns (CA), to general records referring to warm (WWe) or cold (CWe) weather (Figure 2).

The category of rainfall variability considers the indication of rainy years (RY), dry years (DY), certain seasons that were unusual in terms of excess or deficit of precipitation, such as rainy summers (RSm) or dry summers (DSm), or rainy springs (RSp) and dry winters (DW). In addition, flood events (FL) were also identified, such as winter floods (FLW), spring floods (FLSp) or summer floods (FLSm). In some situations, only heavy precipitation (AP) or unusual snowfalls were noted (HS) (Figure 2).

Regarding the category of climate-related natural disasters, they cover a wide spectrum of manifestations, related to social, cultural, geological or biological aspects. They have been divided into two groups: effects on agricultural production and human health. The first group includes locust and mouse invasions (LMI), which destroyed crops and led to weak harvests (WH), ultimately causing high prices and famine (IH); or, on the contrary, in some situations, abundant harvests are mentioned (AH). In addition to these, disastrous effects have also been caused by strong winds (SW), particularly strong storms (SS) often accompanied by hail (HL), which destroyed houses and churches and seriously affected crops. The effects on human health are translated into the appearance of various diseases and epidemics, especially plague epidemics (EP) (Figure 2). Although the action of climate did not generate EP, but due to fundamental biological causes (the bacterium Yersinia pestis), the climate contributed to its spread and to creating a favorable environment for the disease to evolve.

Although the methodology used involves a considerable number of abbreviations, they contribute essentially to the coherent structuring and rigorous interpretation of the identified climatic phenomena. The use of this coded system significantly reduces ambiguities, ensuring a clear and uniform classification of the analyzed events, without leaving room for subjective interpretations regarding the nature of the described phenomena. All abbreviations, together with the





allocation criteria within the climatic items and categories, are systematized and illustrated in detail in Figure 2 and in the abbreviations list at the end of the manuscript. For the years without evidence regarding climatic conditions, the absence of information can be attributed to the fact that, at the time, extreme, special or unusual climatic phenomena were not recorded. Thus, the lack of mentions may reflect normal years or seasons and possible omissions, negligence or documentary losses that occurred over time.

The data obtained were analyzed and interpreted using advanced statistical tools and visualization techniques to comprehensively and rigorously analyze the collected information. In this regard, the R 4.4.1 and MathLab R2024 applications were used to create diagrams and heat maps that facilitated a clear understanding of the variability of climatic phenomena. To present the data in an easy to understand and accessible way, two types of analysis were used, namely neural network analysis (NNA) and social network analysis (SNA), implemented using Gephi 0.10.1 and VennMaker 2.03

software. To analyze the interactions between various climatic phenomena, complex networks were used, which allow the representation and visualization of the relationships that form between variables. In this context, NNA and SNA were applied to model the variability of climatic elements and evaluate their impact, taking into account the non-linear interactions between climatic variables.

*3.3. Data validation and calibration through triangulation and proxy sources*
    A frequently used solution for validating data obtained from historical sources is to relate them to proxy data, which contributes to creating a more objective and complete picture of the past climate. In the present study, proxy databases were used to validate the research results on thermal and pluviometric variabilities. Thus, the reconstruction of solar activity in the 17th century, based on [10]Be from the study by Wu et al. (2018), was used to correlate with the number of phenomena

associated with warm and cold weather in this century, and especially within the MM. At the same time, to reconstruct the thermal regime during the winter, in the context of the influence that the MM had on winters in Europe, data from reconstructions of the $\delta^{18}O$ concentration from the glacier in the Scarisoara cave (Apuseni Mountains - Western Carpathians), in the study by Perșoiu et al. (2017), were used (Figure 2).

    The main comparative source used to validate the historical reconstruction of rainfall variability was the Old World

Drought Atlas (OWDA) maps, which integrate the Palmer Self-Calibrated Drought Severity Index (scPDSI) to quantify the degree of aridity of the climate in Europe and North Africa, starting with 0 AD and until the present (Cook et al., 2015a). For each year of the 17th century, the OWDA maps were processed using the ImageJ analysis platform (v1.54p), through a precise geographical selection of the Transylvanian territory. From each annual map, the color values (RGB code) corresponding to each pixel located in the Transylvanian area were extracted. These codes were subsequently converted into

numerical values of the scPDSI index, based on the chromatic scale used by OWDA. Through this approach, three synthetic indicators were obtained for each year, the minimum, average and maximum value of scPDSI, reflecting the internal variation of the rainfall regime across the entire territory of the principality. The values thus derived were then compared with the frequency and distribution of mentions in historical documents regarding episodes of excess or deficit rainfall, in



order to achieve a validation through convergence between documentary sources and proxy data. The method of obtaining
the values based on OWDA maps can be expressed mathematically as follows:

$$\overline{scPSDI_y} = \frac{1}{n}\sum_{i=1}^{n} scPDSI_{i,y}$$

where $\overline{scPDSI_y}$ is the average value of the scPDSI index within the OWDA maps for Transylvania in year $y$, $n$ is the total
number of analyzed pixels corresponding to the territory of Transylvania, and $scPDSI_{i,y}$ is the scPDSI index value for pixel $i$
in year $y$, obtained by converting the color code (RGB) into a numerical value, reported on the scPDSI index scale.

For greater accuracy, in addition to the data generated based on the OWDA maps, values obtained from $\delta^{13}C$
reconstructions from multiproxy data, by Feurdean et al. (2015), were also used.

## 4. Results

### 4.1. Thermal variability

#### 4.1.1. Unusually cold or warm years and seasons

Regarding the variation of air temperature at the annual and seasonal levels, for the most part, the information
extracted from historical documents refers to cold weather. Thus, 5 years were identified as CY, when very cold weather
prevailed throughout the year, while warm weather, therefore belonging to the HY category, was recorded only in one year.
Many more records refer to cold or warm seasons, most of which were HW, 54 winters during this century being very cold.
In addition, 8 CSp, 4 CA and even 3 CSm were recorded (Figure 3). There are quite a few records about particularly warm
seasons, which go beyond the usual patterns for this climatic zone. Thus, 14 summers are attributed to the WSm category,
with unusually high thermal values, which affected crops, livestock and human society as a whole. In addition to the 55 HWs
reported during the century, there were also exceptionally warm winters, with 12 winters classified as WW (Figure 3). At
least one abnormally cold or warm season was reported in 79 years. 73% of the reports were for unusually cold weather
(reported in 62 years), while the remaining 27% reported exceptionally warm weather (reported in 25 years).

In the case of the records relating to the summer season, these were very disparate at the level of the analyzed
century. In the first half of it, 6 WSm and 2 CSm were recorded, and after a period of 28 years without any records relating
to this season, in the period 1669-1698, in the middle of the MM, there were no less than 8 WSm and one CSm (Figure 4a).
The spring and summer of 1637 were very warm, when *"...a very great heat reigned. Such a heat was never mentioned by
anyone, the grass in the fields was so scorched that even the cattle suffered great shortages"* (Kraus, 1965). The summer of
1607 was "*a summer so hot that even the earth began to burn*" (Réthly, 1962, p.125; Topor, 1964; Rácz, 2001; Dudaș, 2006,
p.33). Very high temperature values were also mentioned in the summer months of 1611, 1617 and 1620 (Réthly, 1962,
1998), and the summer of 1676 was mentioned as being particularly hot and dry.

At the opposite extreme, particularly cold summers record snowfalls, accompanied by frosts that destroyed wheat,
vegetables, fruit trees and forests, as was the case in Transylvania on 7 June 1606 (Réthly, 1962, p. 123; Rácz, 2001). Similar



episodes of severe weather are also recorded between 17 and 20 May 1662, when frosts and snowfalls were reported that affected crops throughout the region (Hanusz, 1890; Millhoffer, 1899; Réthly, 1962), and between 13 and 15 May 1664, frost and hoarfrost phenomena appeared again (Millhoffer, 1899; Stangl & Foelsche, 2022). On May 1, 1665, a heavy

snowfall covered Țara Bârsei, causing significant agricultural damage (Dudaș, 2006, p. 36). Low thermal values during the summer period were also mentioned in 1651, when "*cherries and strawberries ripened late*" in Sighișoara, only in August, and roses bloomed in September (Réthly, 1962, p.180; Stangl, Foelsche, 2022). This plant phenomenology indicates a completely unusual weather, very cold for the summer period within the temperate climate zone, of which Transylvania is part.

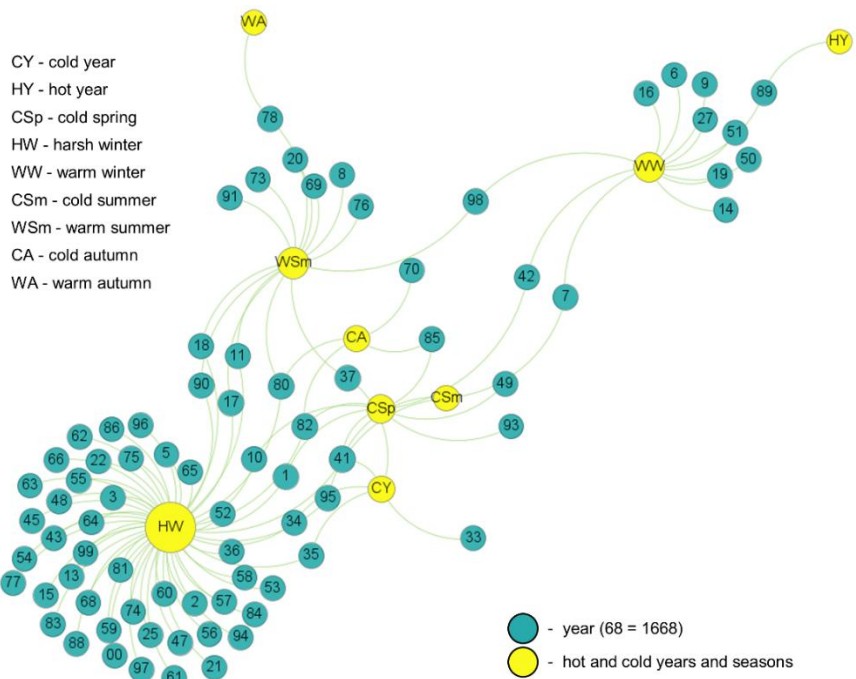

**Figure 3: The temporal network of the thermal regime in the 17th century in Transylvania. Correlations between years/seasons and climatic phenomena associated with air temperature**

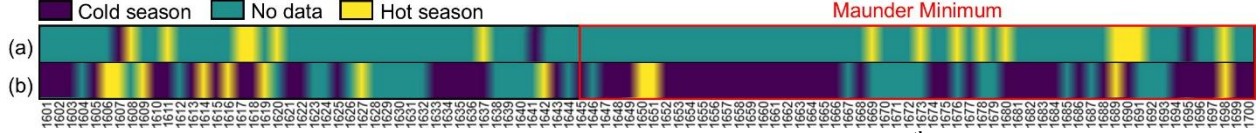

**Figure 4. Temporal reconstruction of temperature-related climatic phenomena in the 17th century on the territory of Transylvania**
*(a – in the summer season; b – in the winter season)*

As for the winter season, the first half of the century is characterized by a great alternation of HW and WW, with a

320 higher frequency of the first category. Thus, until 1645 (at the onset of the MM), 18 HW were recorded, interrupted by 8 WW; especially in the period 1605-1627, the two categories alternated in historical writings. In the second half of the



century, when the cooling was more pronounced, winters were reported as being extremely cold, 36 out of the 55 winters being attributed to the HW category and only three to the WW category. The period 1645-1668 stands out in particular, when out of the total of 23 winters, 19 are reported as HW, 2 WW, and the other two were most likely (due to the lack of information) normal winters, which did not impress with unusual weather (Figure 4b).

An emblematic example of the severity of the winters of the second half of the 17th century is the winter of 1645, described in documents of the time as being exceptionally harsh: "*as no one had ever done before, so that in places many cattle, wild animals and birds thawed. Many large waters and deep lakes froze to the bottom. On the Olt and Mureş rivers, the ice was 8 palms thick, freezing to the bottom in many places, as did the lakes around Sibiu*" (Kraus, 1965). Similar phenomena are also mentioned in the winters of 1647–1648 and 1648–1649, when the snow persisted until April, something completely unusual (Réthly, 1962; Cernovodeanu & Binder, 1993; Dudaș, 2006; Stangl & Foelsche, 2022). The winter of 1659–1660 was also remarkable, not only for its low temperatures, but also for its geopolitical consequences. The Ottoman army, which was supposed to reach Sibiu to support a siege, was forced to stop at Timișoara due to the extreme cold and heavy snow (Guboglu, 1974; Mehmet, 1980; Dudaș & Urdea, 2021). In the same year, severe climatic conditions led the Ottoman army to abandon the siege of Oradea (Roncaglia, 1716). Similar effects of cold on military campaigns are also reported for the years 1662–1664 (Nitri, 1666; Rycaut, 1680) (Figure 3).

The trend of winters becoming colder and more frequent was also evident in the last two decades of the century. Thus, in the penultimate decade, seven out of ten winters were described as extremely cold. The winters of 1680–1681, 1682–1683 and 1683–1684 were characterized by persistent snow and severe frosts that extended into the spring months (Réthly, 1962; Dudaș, 2006; Stangl & Foelsche, 2022). In the last decade of the century, particularly cold winters followed each other almost every year. The winter of 1691–1692 was so severe that the Cris River froze over, allowing the Habsburg army to besiege the fortress of Gyula on ice (Réthly, 1962, 1970). Also, the winters of 1694-1700 were generally extremely cold (except for the winter of 1698) and marked by HS (Dudaș & Urdea, 2021; Stangl & Foelsche, 2022).

However, the climatic cooling specific to the 17th century was not uniform, being interrupted by years and seasons characterized by unusually high thermal values. Chronicles mention mild winters, such as the one in 1650 (Kraus, 1965) or the beginning of the winter of 1698–1699, when "*until January 18, winter was more like summer*" (Dudaș, 2006, p. 38). Also, the summer of 1695, although generally cold, was marked by local episodes of heatwaves, with Turkish chronicles noting that at Şag (Timiş), "*workers and beasts of burden suffered from thirst during the works on the bridge over the Timiş River*" (Guboglu, 1974; Dudaș & Urdea, 2021; Pap, 1822; Millhoffer, 1899; Réthly, 1998). In addition to the summer and winter seasons, springs and autumns characterized by special weather are often mentioned. Historical documents indicate that the 17th century springs were often marked by abnormally low temperatures, late snowfalls and frosts that seriously affected agriculture. Thus, in the spring of 1601, on May 19, there was "*heavy snowfall, cold and frost in Ţara Bârsei*" (Dudaș, 2006, p. 33), and in 1602, on May 28, there were "*heavy snowfalls and low temperatures, as if it were Christmas*" (Réthly, 1962, p. 116). Similar situations are reported in the spring of 1608, with snowfall in Brașov on April 1, and in 1610, when the snow persisted until the beginning of April in Ţara Bârsei (Réthly, 1962).



### 4.1.2. Cold and heat waves

Historical documents often contain notes about cold and heat waves, rather than references to the thermal regime at the annual or seasonal level. Some phenomena are mentioned as having a duration of manifestation ranging from one day to one week, while others extend over a period of 2-3 weeks or even a month. In this century, 64 years, testimonies of these types of climatic phenomena were recorded.

Cold waves were 94 in number (79% of the total of 119 events), recorded within 57 years, and their distribution was uneven throughout the year. Their analysis shows a significantly higher frequency in the spring months (March, April and May, each recording 16 cases) and, partly, in the autumn months (October – 9 cases, November - 9 cases). At the same time, these phenomena, at a rate of approximately 60%, lasted about a month, while in 28.7% of the situations the cold waves lasted between 2 and 3 weeks (Figure 7). Their frequency in the autumn-spring period can be explained by the persistence of cold air masses, generated by the Scandinavian, Greenlandic, East European and even Siberian anticyclones, most often united in winter, which generated long, heavy winters, with heavy snowfalls, short, cold springs and autumns with late and early frosts, frequently reported in chronicles.

In the first half of the 17th century, especially in the third and fourth decades, several winters were marked by extremely low temperatures, serious consequences for the local population and the ongoing military campaigns. Thus, the winter of 1621–1622 is described in historical documents from Cluj as exceptionally cold, with a "*bitter frost*" recorded on February 7 (Réthly, 1962, p. 143; Cernovodeanu & Binder, 1993; Dudaş, 2006). The following winter brought equally harsh conditions, with several sources mentioning that "*several soldiers from the army of Prince Bethlen Gábor died of frostbite while riding*" (Pap, 1822, p. 266; Millhoffer, 1899; Réthly, 1998, p. 1198). Another notable episode occurred on January 16 (1623), during a military crossing of the Tisza River by Prince Rákóczi's army of about 60.000 men. The river, described at the time as swollen, was crossed in extremely dangerous conditions: "*a fierce and savage frost, the like of which had not been seen for 25 years, caused many of the prince's soldiers to freeze and collapse on the spot*" (Kraus, 1959). In 1605, the military campaign led by Ștefan Bocskai, prince of Transylvania, reached a standstill in December, when "*...the cold was so severe that, once it began to snow, it was no longer possible to remain in camp...and the people were dying of hunger, up to 150 soldiers a day...*" (Spontone, 1638).

Heat waves were less frequently recorded in historical documents, with only 25 mentions (21%) in 17 years, evidence that the LIA extended to this part of Europe. Periods of particularly warm weather were recorded in December 1641 and January and February 1642, October-November 1650, and January and February 1696 (Figure 7). Particularly interesting are the heat waves mentioned in October and November 1650, when, due to the heat, the documents indicate that there were flowers everywhere in the fields. Also, the one in February 1683 when the trees budded (Kraus, 1965). Also noteworthy is the year 1673, when sowing was done in January, wheat was harvested in June, and grapes were picked in August (Réthly, 1962). These phenomena are still rare in Transylvania, even during the warming period that began in 1860, and have a lower intensity.




### 4.2. Rainfall variability

*4.2.1. Years and seasons with excess and deficit of precipitation*

From the point of view of the rainfall regime, the 17[th] century seems to have been characterized by excess humidity. Thus, historical documents mention 117 phenomena of deficit or excess precipitation during the 17[th] century, recorded within 74 years (only in 23 years were such phenomena not mentioned). The years and seasons with excess precipitation were in a percentage of 66% (77 mentions within 51 years), while those with deficit were recorded in only 34% of the

situations (40 mentions within 35 years). RY were indicated as being in number of 13, while DY and DT totaled 14 years. Rainy seasons recorded 44 mentions, of which in 19 situations they were HSW, in 12 RSm, in 7 RA and 6 Rsp. At the same time, 20 events associated with FL were mentioned, which manifested themselves at the level of an entire year (12) or characterized certain seasons (FLSp – 3; FLSm – 4; FLW – 1). Regarding the seasons with moisture deficit, these were 26 in number, the most frequent being DSm with 17 mentions, followed by DW with 5 mentions, DS with 3 mentions and DW

with one (Figure 5).

The links that form between the items in Figure 5 indicate that most HSW events were related to HW, so in the 17[th] century winters in Transylvania were excessively cold and with abundant precipitation in the form of snow, the resulting snow cover in many cases persisting for a long period of time. In most cases where WW or WSm are reported, they are associated with mentions of FL, FLSp or FLSm, which indicates that the respective floods may have been determined by the

rapid melting of snow accumulated during the winter or due to the overlap of snowmelt with new liquid precipitation specific to spring, as well as due to convective precipitation during the summer. In the case of years and seasons characterized by FL, the distribution at the century level is uneven, with 10 such phenomena recorded in both the first and second half of the century. As for the rainy or dry years and seasons, these are also irregularly distributed, without being able to identify a prolonged period of moisture deficit or excess. In the second half of the century, there were more records of

excess moisture (41 records), compared to the first half, which is characterized by 36 such records, but also more testimonies of moisture deficit (22), compared to the first half in which 18 were recorded (Figure 6).



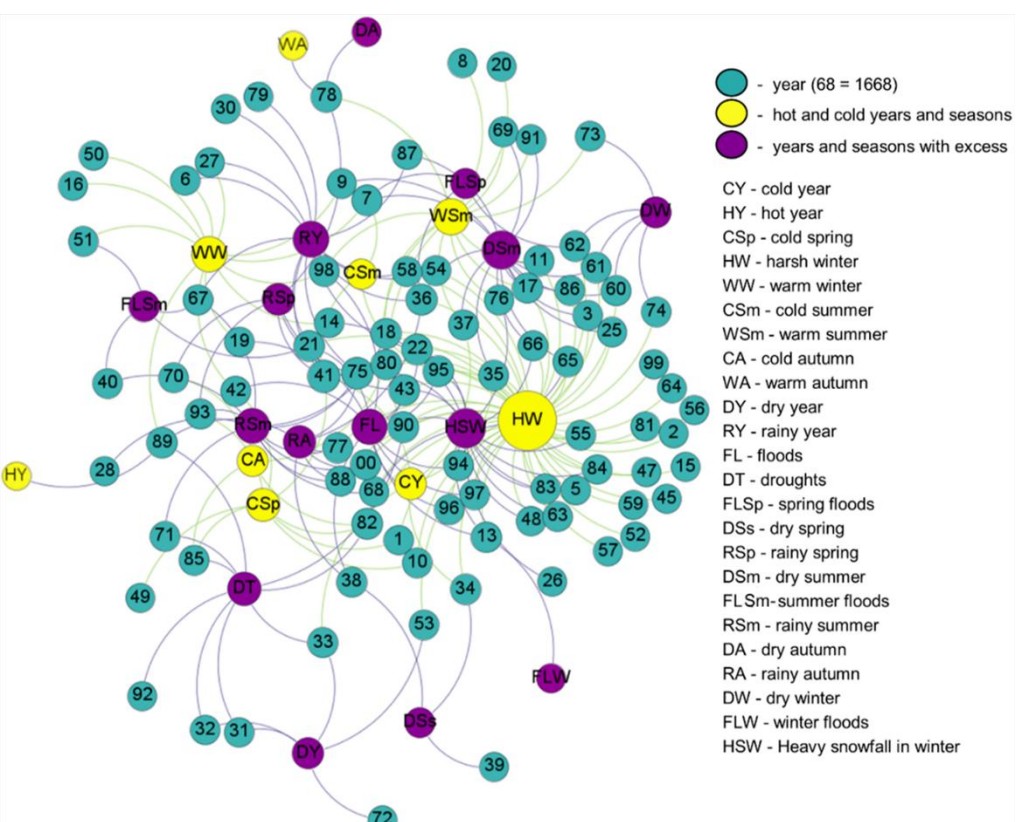

**Figure 5: The temporal network of the thermal and pluviometric regime in the 17th century in Transylvania. Correlations between years/seasons and climatic phenomena associated with air temperature and atmospheric precipitation**

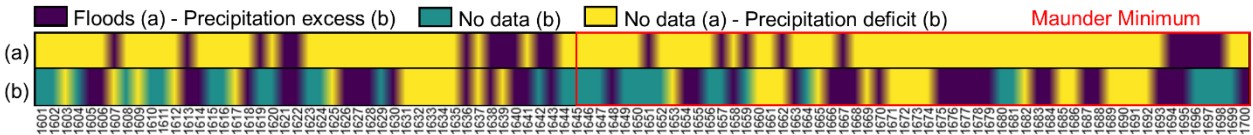

**Figure 6: Temporal reconstruction of climatic events related to excess and deficit of precipitation in the 17th century on the territory of Transylvania** *(a – floods; b – excess/deficit of precipitation)*

Heavy rains were recorded in 1627 (Dudaș, 2006), in the summer of 1628 (Réthly, 1962; Cernovodeanu & Binder, 1993) and in 1630, when in Mediaș "*the wine came out in abundance, but sour*" (Réthly, 1962, p. 150; Cernovodeanu & Binder, 1993). Drought struck in 1625, drying up the grapes and even the mushrooms on the trees in Țara Bârsei (Topor, 1964; Dudaș, 2006, p. 34; Teodoreanu, 2017). The early 1640s, one of the wettest decades, registered major floods in July 1641 (Réthly, 1962; Dudaș, 2006) and continuous rains in Alba Iulia in late 1641 and early 1642 (Réthly, 1962, p. 165; Rácz, 2001). In 1675, persistent rainfall from spring to autumn led to poor harvests (Réthly, 1962; Dudaș, 2006), and in Mediaș "*little and inferior wine*" was produced due to summer rains (Réthly, 1962). The last decades of the century showed the most intense variability. The winters of 1682–1685 brought heavy snowfall and precipitation (Réthly, 1962), while in March 1684 there were floods and thick mud (Réthly, 1962). Torrential rains hit Turda in the summer of 1687, and in 1688



there were heavy precipitations in both summer and autumn (Réthly, 1962; Dudaş, 2006). In 1689, a poor grape harvest was again reported in Mediaş due to "*the heavy rains in the autumn of 1689*" (Réthly, 1962, p. 237).

Frequent AP and SS generated in 1621 "*great floods, which gave rise to and caused a great plague in Transylvania and Hungary*" (Kraus, 1959). A year later, in 1622, "*there were great floods in Transylvania, causing great damage. Then followed a great famine and a plague*" (Kraus, 1959). Also in 1622, Kraus (1959) indicated that "*in November, as well as in*
*the previous month, following the continuous rains and the breaking of clouds, the ground became so loose that it could be said that everything was floating on water. The roads throughout Transylvania were bad and especially through villages and marshy places almost no one could travel without harnessing many good horses and oxen. ... All this, as well as the collapse, in many places, of many hills and mountains, were signs of doom*".

Drought was a recurring phenomenon in 17th-century Transylvania, especially in certain decades. The first
mentions appear in 1603 and 1607, "*when the grass caught fire in the fields and the forests burned*" (Dudaş, 2006, p. 33; Teodoreanu, 2017) and in 1609 (Réthly, 1962). Other episodes are recorded in 1611, 1614, 1617 and especially in the autumn of 1618, which strongly affected the harvest (Topor, 1964; Dudaş, 2006, p. 34). The 1630s were the driest, with droughts in 1631–1632 (Réthly, 1962), in 1633, "*when almost everything dried up in Ţara Bârsei*" (Dudaş, 2006, p. 34), as well as in 1634–1636 and 1639 (Millhoffer, 1899; Réthly, 1962; Rácz, 2001). In 1669, no precipitation was recorded
between 6 August and 6 November (Topor, 1964, p. 16; Stangl & Foelsche, 2022). The 1670s continued with periods of drought, especially in 1672, 1673–1674 and 1678, when "*the proudest forests in Ţara Bârsei burned down*" (Topor, 1964, p. 16). In 1686, drought and fear of a possible famine were mentioned in Timişoara (Cernovodeanu & Binder, 1993; Dudaş & Urdea, 2021).

### 4.2.2. Episodic manifestations of excess and deficit of precipitation

Historical documents also mention 171 episodic manifestations of excess or deficit precipitation within 66 years. Of these, 61% (104 events) were phenomena characterized by excess precipitation, recorded within 49 years, while only 39% (67 events) were associated with deficit precipitation, occurring within only 38 years (Figure 7).

The most frequently mentioned were FL (67 phenomena), of which 47 are generalized at the monthly level, 17
manifesting between 2 and 3 weeks, and 3 extending over 1-2 days. No less than 44 of these manifested themselves in May-August, with 12 mentions of FL in June and 20 in July, at the opposite pole being February and September (Figure 7).

In May 1682, major rivers overflowed their banks (Réthly, 1962; Dudaş, 2025), and in March 1689, catastrophic floods followed the sudden thawing of the heavy winter snows (Réthly, 1962). Similar events were also reported on 10 May 1635, in Mureş, in Braşov in June–July 1635, and throughout Transylvania in November 1635. Autumn rains and floods
were recorded in December 1636, with more flooding in Cluj on 13 January 1638 (Réthly, 1962). Catastrophic climatic phenomena include the floods caused by snow melting on the Crişul Negru in April 1659 (Dudaş, 2006) and the severe floods in Sibiu on July 25, 1659 (Réthly, 1962). An 1639 account states that: "*On May 23, heavy rains fell throughout*





*Transylvania, so that all the rivers in the country overflowed their banks and many people and livestock perished*" (Kraus, 1959), suggesting the magnitude of the events associated with excess precipitation.


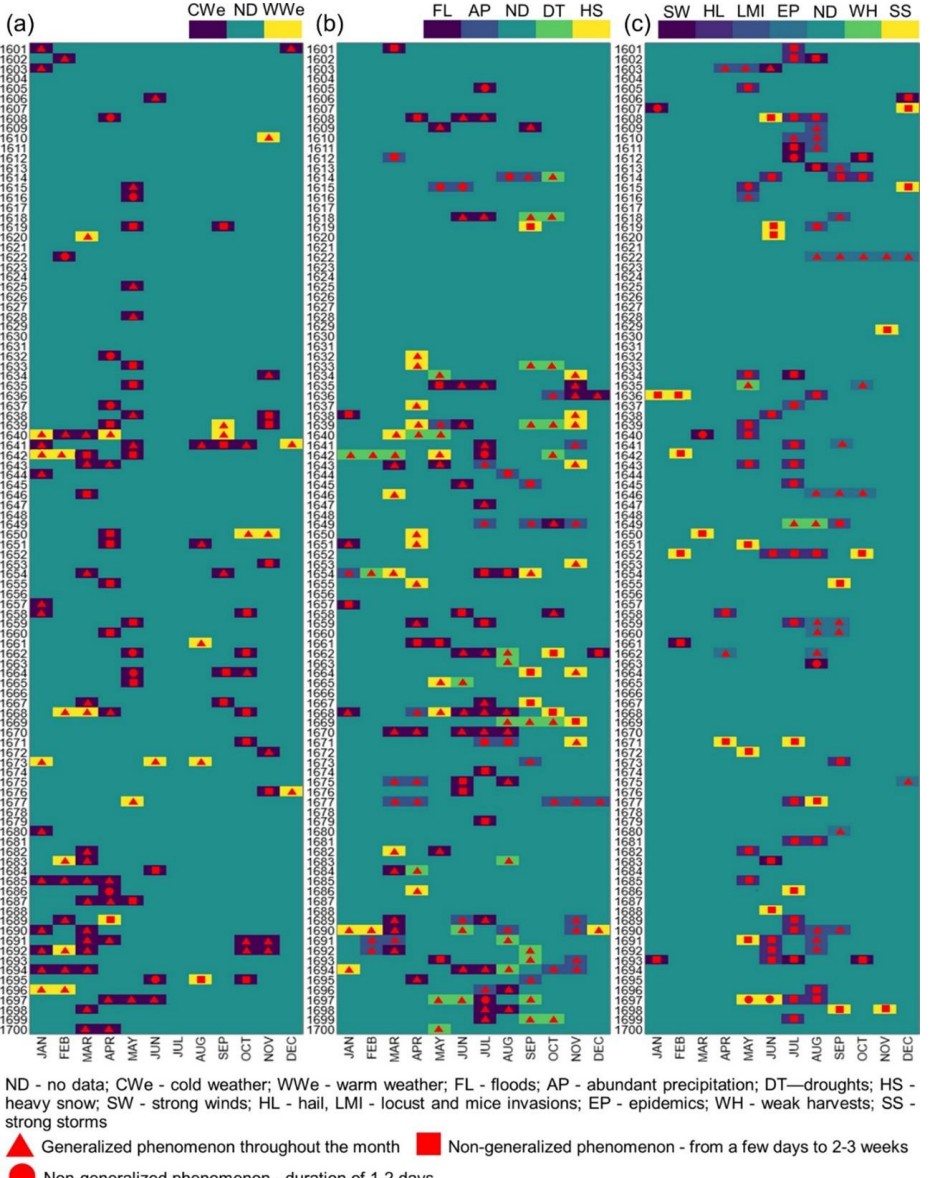

ND - no data; CWe - cold weather; WWe - warm weather; FL - floods; AP - abundant precipitation; DT—droughts; HS - heavy snow; SW - strong winds; HL - hail, LMI - locust and mice invasions; EP - epidemics; WH - weak harvests; SS - strong storms

▲ Generalized phenomenon throughout the month ▮ Non-generalized phenomenon - from a few days to 2-3 weeks

● Non-generalized phenomenon - duration of 1-2 days

**Figure 7: Cold and heat waves (a), episodic manifestations of excess and deficit precipitation (b) and other risky climatic phenomena (c)**

Often associated with FL, AP were recorded in 37 cases, most of which (29 events – 78.4%) were generalized at the monthly level and only 5 occurred over 2-3 weeks (Figure 7). November was mentioned 7 times as extremely rainy,





followed by September with 6 mentions and March, July and August with 4 mentions. On 2 June 1640, a violent storm hit Sag, causing extensive damage: "*...a cloudburst occurred which caused great damage, many people and cattle perishing*" (Kraus, 1959). Later that year, on September 14, 1649, coinciding with the feast of the Exaltation of the Holy Cross, a series

of SS caused widespread flooding: "*...unbelievably high waters came throughout Transylvania, causing significant damage to houses, people and livestock*" (Kraus, 1959).

DT is individualized by the presence of 34 phenomena throughout the century, all highlighting particularly dry months. September was mentioned most often as being dry, with 8 entries, followed by October (7 mentions) and August with 6 mentions. Severe drought is documented in September 1697, affecting the Ottoman troops stationed in Timișoara

(Dudaș & Urdea, 2021), as well as in the period September–October 1699 and in May 1700, when it is reported in Țara Bârsei (Topor, 1964; Teodoreanu, 2017).

HS are mentioned 33 times, of which 27 mention the whole month as affected by snowfall, while 6 records refer to 2 to 3 weeks. Unusual snowfalls, which attracted people's attention, were recorded in April (8 years), November (8 years), March and September (both 4 years). In some years (3) snowfall was recorded even in May (Figure 7). Between 13 and 15

May 1664, HS were recorded, which led to significant crop damage (Millhoffer, 1899; Réthly, 1962; Stangl & Foelsche, 2022). One account indicated that: "*In March an incredibly heavy snowfall fell, which remained on the fields throughout Lent, until Easter (6 weeks), causing great damage to wheat and other crops*" (Kraus, 1959).

### 4.2.3. Special meteorological phenomena

In the 17th century, certain special meteorological phenomena are widely mentioned in the territory of Transylvania, with localized action and limited in time, but with potential disastrous effects on local communities. Thus, historical documents indicate 39 events related to particularly dangerous HL, consistent within 25 years, which caused significant losses in agriculture, crop destruction and famine (Figure 7). Most were recorded in the warm months of the year, with a peak in July (14 records), May (8 records) and August (8 records). Many mentions also refer to SS (25 records within 20

years), distributed uniformly throughout the year, but with a peak in June (5) and May (4). SW are also represented by 16 records in 13 years, most being in July (4) and August (3), but they were also encountered during the winter period (4).

Among the most relevant HL episodes are the one on May 16, 1605 in Sibiu (Réthly, 1962), December 3, 1607 in Brașov and Mediaș (Réthly, 1962; Dudaș, 2006), June–July 1608 with multiple hailstorms in Sibiu, Sighișoara and in Țara Bârsei (Dudaș, 2006) and June 28, 1614 in Mediaș (Réthly, 1962). Such phenomena were also reported on May 22, 1634 in

Sibiu when "*all the fruits, vineyards and orchards perished. The hail was the size of walnuts, and the largest grains were the size of chicken eggs. The layer of hail reached everywhere up to the knees, and in places where the sun did not shine, the hail remained for eight days in a row*" (Kraus, 1959).

Transylvania was repeatedly affected by SW and SS, often accompanied by significant material damage (Figure 7). Among the earliest mentions are the storm at Alba Iulia on 16 August 1602 and the one in June 1603 at Cluj (Réthly, 1962).

In the winter of 1606–1607, two consecutive episodes of strong winds are reported on 29 December 1606 and 1–2 January





1607 at Brașov, followed by a strong storm on 3 December 1607 at Brașov and Mediaș (Réthly, 1962; Dudaș, 2006). Also, the date of July 13, 1634 remained in the history of Transylvania through the storm of great intensity, which simultaneously affected several localities and "*broke many windows, tore roofs, as well as large trees from the ground. Everywhere the buildings suffered great damage*" (Réthly, 1962). In 1336, on February 7 "*there was a terrifying storm in Sighișoara and its*
*surroundings...on the same day, a large hailstone fell accompanied by thunder and lightning. In three places in the city it thundered*" (Kraus, 1959). In 1643, following a SS, lightning struck the Ghimbav fortress in Țara Bârsei, and the locality burned to the ground (Kraus, 1959).

### *4.3. Climate-related natural disasters*

### *4.3.1. Effects on agricultural production*

The effects of climate on agricultural production were significant during the analyzed century, because the society depended on an agriculture that produced only for immediate needs (subsistence) which was prone to being affected by unfavorable climatic conditions. Thus, historical documents mention 11 years in which disastrous effects on agriculture caused by LMI were recorded. Their effects (but not only) were translated into 20 years of WH and 26 years with IH. IH
manifested itself throughout the 17$^{th}$ century, from which the periods 1601-1606 and 1648-1652 were individualized as intervals with generalized IH, which determined diseases and malnutrition (Figure 9b). These events seriously affected society, making people much more prone to diseases, which favored the rapid spread of epidemics. Besides the 20 years with WH, only 5 years with AH were mentioned (Figure 8 and 9). The harvests were probably average, but sufficient for subsistence in the years without any mentions.




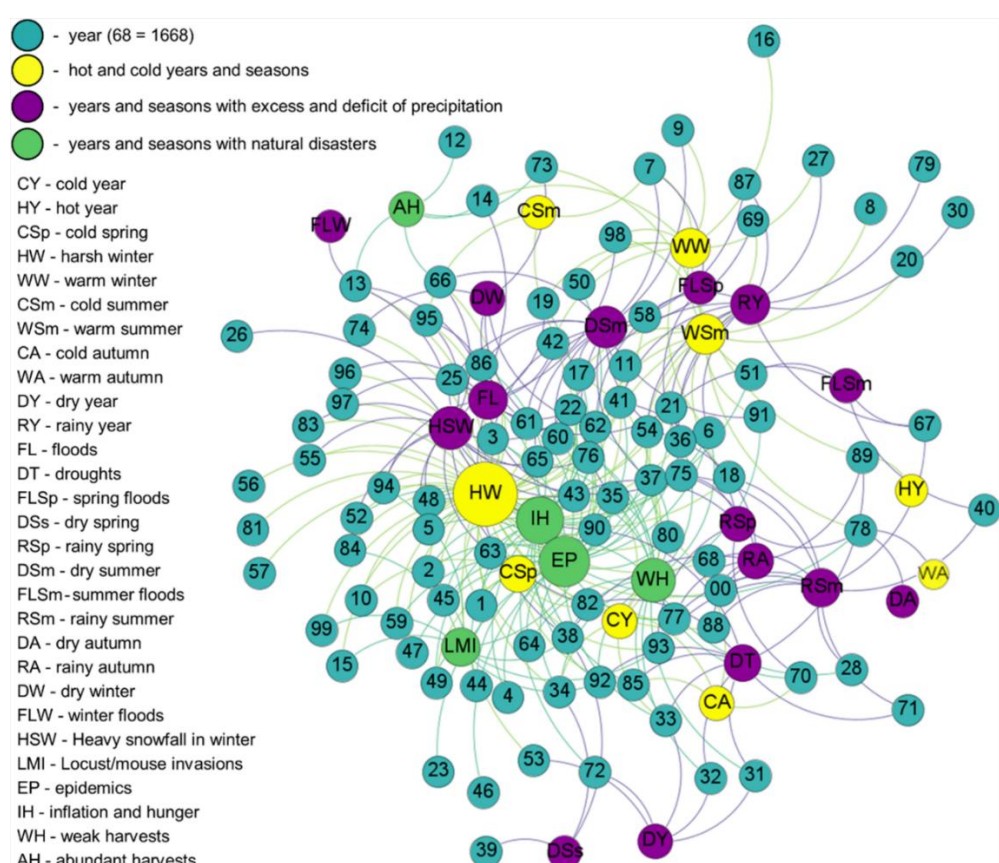

**Figure 8: The temporal network of the thermal and pluviometric regime, respectively of the natural calamities associated with the climate in the 17th century on the territory of Transylvania. Correlations between years/seasons and climatic phenomena related to air temperature, atmospheric precipitation and climatic risk phenomena**


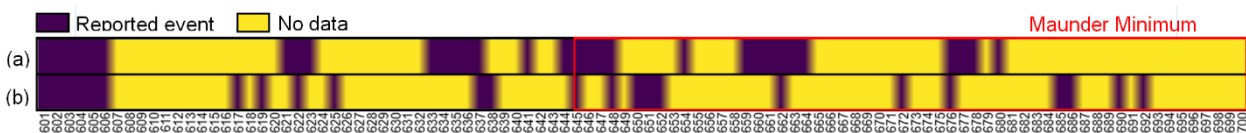

**Figure 9: Temporal reconstruction of the occurrence of climate-related risk phenomena in the 17th century on the territory of Transylvania *(a – plague; b – famine and famine)***

Overwhelmingly, LMI, WH and IH were recorded in the same years as HW, CSp, CY and many cold wave events, as well as FL and HSW. This indicates that crops were most likely destroyed by unusually cold weather (especially in spring) and excess precipitation associated with FL and AP. In some situations, crops also appear to have been affected by prolonged DT, especially during periods when DT set in after cold and wet springs, severely affecting subsequent plant development and compromising yields (Figure 8). In 1635, grain suffered greatly due to DT, as did wine, which "*came to*

*cost 75 florins in Sibiu. The lack of fodder led to the decimation of many cattle, and the absence of meat and grain generated famine and its epidemiological corollary, the plague*" (Georgita, 2022). Months in which LMI was extremely prominent were especially August (7 mentions), September (4 mentions) and April, May and July (all with 3 reports each). In





particular, the months of April-May of 1603 were recorded as being characterized by severe food shortages, as were the periods of August-September 1690 and July-August 1610 (Figure 7). The year 1638 was mentioned in historical documents

as a year with LMI, when "*towards autumn, a great multitude of locusts came and settled like a great cloud, so that the Sun became dark*", and people tried to chase them away by shooting with rifles and with various noisy tools, with fire and burning straw (Kraus, 1959). The source also mentions that "*...these bugs were a punishment sent by the Lord and were a sign foreshadowing the Turkish and Tatar enemy invasions that followed...*" (Kraus, 1958); the enemies invading (like the locusts) from Țara Bârsei in 1658.

Generated by the LMI, but also by the adverse weather, the IH seriously affected Transylvanian society. Historical sources mention the year 1601 as being seriously affected by the IH, so that "*people started eating dogs and cats, sometimes even reaching the point of cannibalism*", and "*..cat meat was sold at the market for 8 crăiţari, half a kilo. The remaining food was consumed uncooked due to the lack of wood*" (Georgita, 2022). At the same time, in June and July 1602, in collaboration with the EP, the terrible famine led some people (especially children) to resort to extreme measures, as some
sources mention that "*the hunger was so great that the poor children ate their own dying parents*" (Georgita, 2022).

### 4.3.2. Effects on health and society

The connections that form between different variables in Figure 8 indicate a very good collaboration between the episodes of IH, LMI, WH and EP; the vast majority of these variables being reported together within the same year (Figure
8). Climate had a particular influence through its effects on agriculture, transposed into WH and IH, events that weakened the nutritional status of the population, making it more vulnerable to diseases and favoring the spread of epidemics. Thus, climate played an indirect, but essential role in amplifying the impact of epidemics, especially EP, in 17th-century Transylvania.

EP were mentioned within 31 individual years, generating a high mortality among the resident population (Figure
8). From the analysis of Figure 9a, it can be observed that certain epidemics are individualized as multiannual. Thus, the periods between 1601 and 1606, and 1659 and 1664, respectively, can be mentioned, when the plague is mentioned in Transylvania for a period of 6 consecutive years. In the intervals 1633-1637 and 1644-1648, 5 consecutive years with EP were recorded in the principality. In the first half of the century, the mentions are more numerous (20), compared to the second, which is individualized by 11 years with EP (Figure 9). At the same time, very serious episodes of EP are associated
with the months of August-December (16 events) (Figure 7); Worth mentioning are the periods August-December 1622 and August-October 1646, when the outbreak of the plague coincided with frequent reports of severe floods, heavy rains and extreme atmospheric and pedological humidity conditions, suggesting a possible triggering role of hydrometeorological factors in the worsening of the epidemic context (Figure 7).

In some situations, historical documents indicate with certainty the causes of the outbreak of epidemics, as in the
case of 1601, when "*the spread of the plague within the city walls was certainly favored by the population crowding, the lack of water and food, the excessive increase in the price of food, as well as the poor sanitary condition*" (Georgita, 2022).





In 1621, the EP seems to have been caused by "*great floods and storms*" (Kraus, 1959), and between 1633-1636 "*climatic vagaries and famine triggered and sustained another plague epidemic*". In the summer of 1634 "*the plague made its presence felt again in Transylvania, but especially in the Saxon cities, being accompanied by great climatic imbalances*"
(Georgita, 2022). Disastrous effects on society caused by the EP were mentioned in 1654, when Saxon chronicles indicate that people "*saved themselves by fleeing over the mountains to the other side of the Danube*" (Georgita, 2022). In 1660-1661, according to some estimates, the plague killed "*half of the population of this city*" in Sibiu (Georgita, 2022), and in 1633, in Brasov, no less than 11.000 people died between June 11 and December 1st (Kraus, 1959).

**5. Discussion**

Compared to the 16th century, analyzed in the previous study by Gaceu et al. (2025), which was marked, in Transylvania, by long periods of relatively warm weather and numerous mentions of warm years and summers, or even unusually mild winters, the 17th century is distinguished by a high frequency of events associated with cold weather, especially particularly cold winters (54 mentions of HW). This is even more evident after the onset of the MM (1645) when
the number of reports regarding HW increased exponentially, with 36 HW and only 4 WW recorded out of a total of 55, while in the previous century only eight HW were reported. These mentions are in agreement with events across the European continent, where years with extremely cold winters also prevailed, as attested by data (Riedwyl et al., 2008b) and historical documents (Pfister et al., 1994; Stangl and Foelsche, 2022).

The most plausible explanation for the high frequency of very low temperatures during the 17th century is low solar
activity. Solar activity during the LIA was fluctuating, but was generally characterized by low sunspot numbers (Wang et al., 2000; Usoskin et al., 2015; Owens et al., 2017). After the MWP, solar activity began to decline, culminating in the MM (1645–1715), when sunspots were extremely rare, sometimes completely absent (Eddy, 1977; Ribes & Nesme-Ribes, 1993). According to the decadal reconstruction by Wu et al. (2018), the average number of sunspots steadily decreased in the 17th century, from 35 in the first half to only 8 in the second, and the period 1645–1695 had an average of only 6 spots (Wu et al.,
2018). In the present study, the MM period is characterized by an exponential increase in the number of records associated with a low temperature regime. Thus, 122 events associated with cold weather were mentioned; in contrast to only 36 events accumulated in the rest of the century. During this period, the reports regarding the number of particularly severe winters also increased exponentially, 36 winters out of a total of 55 being particularly cold. Figure 10 is revealing in this regard, the continuous decrease of the line indicating that solar activity is associated with the increase in the number of severe cold
reports. This is also indicated for the region by Stangl and Foelsche (2022), and the studies of Lookwood et al. (2010) and Luterbacher (2001) attest that the reports of cold weather also increased in the rest of Europe during the low solar activity of the MM period.



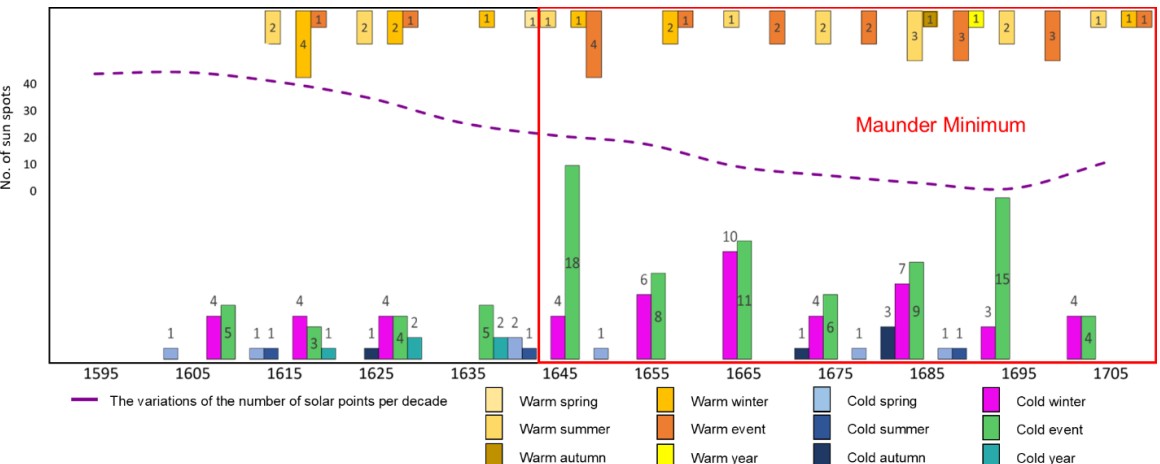


**Figure 10: Correlation between the average number of sunspots per decade and the years/seasons/phenomena associated with the temperature regime in Transylvania in the 17th century based on reconstructions from historical documents (Data source: Wu et al., 2018)**

The vast majority of events captured by historical documents analyzed in Transylvania refer mainly to particularly cold winters, with records of anomalies in other seasons being quite limited. For reconstructing the temperature regime in the winter months from proxy data, oxygen-18 ($\delta^{18}O$) isotopes from glaciers represent one of the most efficient methods, given that tree growth rings reflect temperatures during the active vegetation season, which in temperate regions corresponds mainly to summer and partly to spring or autumn, the ability of these data to indicate the winter air temperature regime being limited (Bolzan and Pohjola, 2000; Huang et al., 2019). Thus, a study by Perșoiu et al. (2017) aims to reconstruct the winter climate variability in Transylvania based on $\delta^{18}O$ analyses from ice cores taken from the Scărișoara Cave (Apuseni Mountains, Romania). The authors report a sharp decrease in winter temperature starting ~800 years ago, with a minimum during the MM and LIA, when $\delta^{18}O$ values reached their lowest levels, accompanied by a D-excess below 6‰, the lowest value in the last approximately 700 years. The reconstructions of the amount of $\delta^{18}O$ partially match the data obtained from historical sources and indicated in Figure 11, taking into account the fact that a lower amount of $\delta^{18}O$ indicates a HW, and the periods of increase are related to WW. This is very evident in the period after the onset of the MM, especially between 1654 and 1665, as well as between 1674 and 1690, when the amount of $\delta^{18}O$ was on a downward curve and a very large number of HW were reported (Figure 8). In the period 1610-1625, 3 WW (but also 9 HW) are mentioned against the background of a decrease in the amount of $\delta^{18}O$ in the glacier, which may indicate some inconsistencies (Figure 11). The discrepancies between low $\delta^{18}O$ values and the occurrence of WWs can be explained by a combination of climatic and local factors. Studies show that $\delta^{18}O$ values are influenced not only by temperature, but also by variations in the path and source of air masses (Holme et al., 2019). Also, studies on the relationship between $\delta^{18}O$ and NAO indicate that in Central and Western Europe, milder winters can coincide with low $\delta^{18}O$ values due to changes in the trajectory of westerly winds and cyclones (Baldini, 2008). Also, in two of the three cases, the reporting points of those winters were in the western part of Transylvania, on the border with Hungary, at a significant distance from the Scarisoara Glacier and in a plain climate. In this




regard, the study by Nagavciuc et al. (2020) shows that $\delta^{18}O$ values in the Carpathian Mountains are often lower than in plain regions, due to altitude and regional atmospheric circulation.

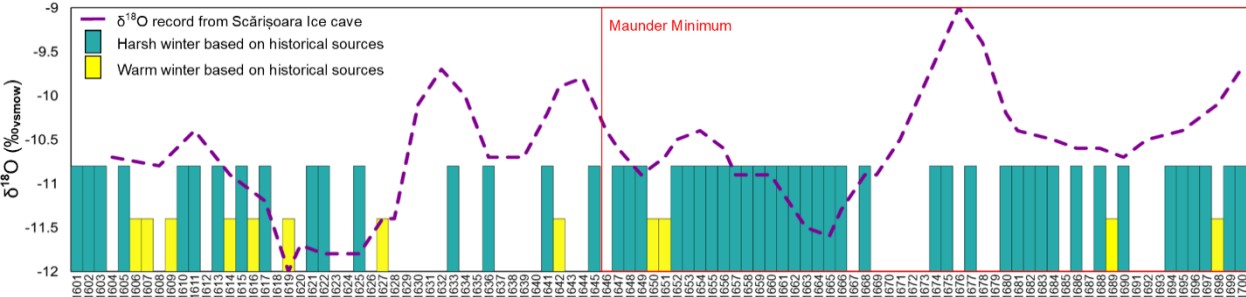

**Figure 11: Correlation between δ18O variations in the Scarisoara Cave glacier (proxy data for winter temperature reconstruction)**
**and the description of 17th-century winters in Transylvania based on reconstructions from historical documents** *(Data source: Perșoiu et al., 2017)*

The 17th century is also characterized by a higher frequency of events related to excess precipitation compared to the previous century, which was dominated by droughts and precipitation deficits (Gaceu et al., 2025). For the rest of
Europe, studies (Pauling and Paeth, 2006; Dieppois et al., 2013; Huo et al., 2021) indicate that the 17th century was marked by an increase in precipitation variability, with alternations between excessively rainy periods and severe drought episodes. In general, Europe was characterized by colder and drier winters and summers and autumns with abundant precipitation. According to some authors (Perșoiu and Perșoiu, 2018), the wetter local conditions during the LIA, starting with the 17th century on the territory of Transylvania, were the result of a combination of the intensification of local cyclonic activity and
the increase in moisture transport from the North Atlantic, associated with a more southerly positioning of the westerly winds, characteristic of the NAO.

Although the LIA is often characterized as a dry period, Feurdean et al. (2015) show, based on multi-proxy data reconstructions from the northern Carpathian Mountains, that in the territory of present-day Romania, a peak in precipitation was recorded during the MM (a peak that was surpassed only by the end of the LIA, in the period 1800-1850). At the same
time, the study by Drăgușin et al. (2014) analyzing the stable isotopes of $\delta^{18}O$ from calcite deposited in stalagmites from different caves in the Carpathian Mountains and the Balkan Mountains during the Holocene confirms that in the second half of the 17th century the climate would have been colder and wetter in winter. Moreover, this period seems to have been the rainiest of the last approximately 300 years in this area. Some authors indicate that these aspects can be explained by the reduced solar radiation during the MM period, which can influence the regional climate by modifying the general
atmospheric circulation (Ineson et al., 2011; Vita-Finzi, 2011; Van Dorland et al., 2006).

The data from these natural reconstructions fit perfectly with the data obtained in this study from historical documents. The latter show a significant increase in the number of reports of events and periods associated with excess precipitation. If by 1640, when the $\delta^{13}C$ curve begins to increase, only 46 events (0.9 events on average per year) with excess





precipitation were reported, in the following years no less than 140 such events (2.3 events on average per year) were
mentioned (Figure 12).

This is also highlighted by the scPDSI values on the OWDA maps (Cook et al., 2015b). Figure 12 highlights the
correlation between the minimum, maximum and average values of the scPDSI index in the Transylvanian area and the
number of reports associated with excess and deficit precipitation. Values close to +6 of the scPDSI index in Transylvania
corresponded to frequent mentions of excess precipitation, especially in the second half of the century. In contrast, averages
close to -6 were associated with extended periods of drought throughout the century (Figure 10).

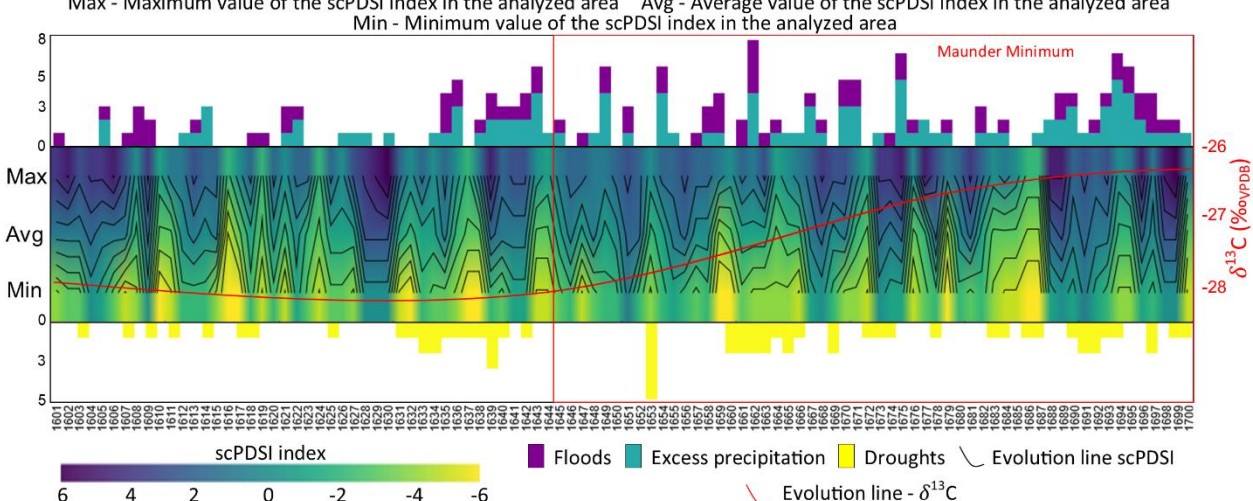

**Figure 12: Correlation between the values recorded on OWDA maps and the years with excess/deficit precipitation in the 17ᵗʰ**
**century in Transylvania based on reconstructions from historical documents** *(Data source: OWDA - Cook et al., 2015a; PAGES2k*
*Consortium, 2017; δ13C reconstruction - Feurdean et al., 2015)*

The data on the OWDA maps and those in historical documents match to a large extent, but not completely. The
discrepancies result at first glance from the fragmentation of the Transylvanian relief, with precipitation being abundant in
western Transylvania compared to its center, due to the influence of the western atmospheric circulation and the proximity to
the humid fronts sent by the Atlantic and Mediterranean cyclones. At the same time, the Apuseni Mountains act as an
orographic barrier, causing the condensation of humid air and the increase in precipitation on the western slope. Therefore, a
report of excess precipitation localized in western Transylvania does not always apply to the entire province, because in its
center drought could prevail to certain extent.

In the climatological analysis of the 17ᵗʰ century, the lack of climate information for certain periods can be
explained by the fact that people recorded only exceptional phenomena, which may indicate a stable climate in the years
without records (Guidoboni, 1998). Also, limited access to education, especially in small and rural communities, may
explain the absence of documentary information. Thus, the lack of sources does not necessarily reflect the absence of climate
events, but the priorities and recording possibilities of the century (Pfister, 1999).





## 6. Conclusions

The 17th century in Europe was marked by multiple crises, both social and economic, demographic and political. To all this was added the great climatic variability, the continent experiencing the effects of the LIA during this period, and at the end of the century the coldest period in the last at least 1000 years in Europe, namely the MM, set in. The effects of the LIA, respectively the MM, were also felt strongly in Transylvania, especially through a high frequency of cold weather phenomena, in 79 years there were mentions of particularly cold weather (73% of the total records). There were also 94 cold waves in 57 years, respectively 54 winters were categorized as HW in this century. However, historical sources indicate thermal values in the summer that remain high, which suggests a delay in cooling in the east of the continent compared to the central and western regions. Such events occurred over a 25-year period, but accounted for only 27% of all mentions. These events were driven by reduced solar activity and the NAO-action, which allowed more frequent intrusion of cold air masses from northern Europe and even Asia, favoring colder and more persistent winters. In addition, the 17th century in Transylvania was characterized by a positive precipitation regime, with events associated with excess precipitation being 4.2 times more than those with deficit (186 seasons compared to only 44); 51 years and seasons were marked by phenomena associated with excess precipitation, only 35 being deficient from this point of view. This is felt more strongly during the MM period, when precipitation amounts were the highest, and 44 out of 55 years (80%) recorded excess precipitation or periodic manifestations thereof. The abundant precipitation could also have been determined by the reduced solar activity, which influenced the atmospheric circulation at continental level, favoring the penetration of wetter and colder air masses.

The unusually cold and wet climate, sometimes causing severe droughts, had a profound impact on agriculture in Transylvania. Historical documents attest to numerous cases of IH and WH, determined by capricious weather and sometimes by frequent LMI (32 years with such phenomena). These economic and climatic realities favored the outbreak of social conflicts, aggravated by the repeated outbreak of EP, especially in the second half of the century. EP episodes were determined by a combination of climatic, social, economic and sanitary factors, which created an ideal context for the emergence and spread of diseases. In turn, these generated massive demographic losses, local economic collapse and a social climate marked by fear and instability.

Finally, proxy data provide an independent validation of qualitative historical sources, confirming, in general terms, the presence of severe climatic conditions and pronounced climatic variability. The climate of Transylvania in the 17th century did not undergo fundamental transformations, continuing to remain within the temperate-continental pattern specific to the region and present today, but this cannot be characterized as constant, as it was marked by significant variability, manifested by a general cooling trend. The correlation of multiple evidence supports the hypothesis that this period was characterized by climatic stress in the context of the LIA, respectively the MM which generated high social vulnerability, with a significant impact on human communities in Transylvania.



## 7. Study limitations

Information on the climate of Transylvania extracted from historical documents represents valuable evidence for its reconstruction. However, the study suffers from certain methodological and documentary limitations. First of all, the documentary coverage is incomplete, given the low literacy of the population of those times and the inaccessibility of certain writings, which determines a limited volume of historical information for this century. At the same time, the study is not exhaustive, so it did not analyze all the writings of the time, all the authors and all the accounts, but we believe that adding them would not significantly change the general picture of the climate of the 17th century. In addition, historical data have an inherent degree of subjectivity, being estimates of climatic conditions based on personal experiences. At the same time, in some situations, the nature of the information does not allow for generalization at the year or season level, considering that they may refer to climatic phenomena that lasted for a limited period of time and had a local character, so this information cannot be extrapolated, which made the research difficult and fragmented.

**Data availability statement**
The raw data supporting the conclusions of this article will be made available by the authors, without undue reservation.

**Author contributions**
Conceptualization: OG, TC, ȘB. Data curation: TC, FD, SM, MM, CM. Formal analysis: TC, FD. Methodology: OG, TC, ȘB. Software: MM, TC, CM. Visualization: FD, OG. Writing (original draft preparation): OG, TC, FD. Writing (reviewing and editing): ȘB, MS, MM, CM.

**Acknowledgments**
This research has been supported by the University of Oradea, Romania.

**Competing interests**
The authors declare that they have no conflict of interest.

**Abbreviations**
LIA - Little Ice Age
MM - Maunder Minimum
MWP - Medieval Warm Period
NAO - North Atlantic Oscillation
NAO+ - North Atlantic Oscillation positive phase
NAO- - North Atlantic Oscillation negative phase
CY - cold year
HY - hot year
CSp - cold spring
HW - harsh winter
WW - warm winter
CSm - cold summer
WSm - warm summer
CA - cold autumn
WA - warm autumn
DY - dry year
RY - rainy year
FL - floods
DT - droughts
FLSp - spring floods



DSs - dry spring
RSp - rainy spring
DSm - dry summer
FLSm - summer floods
RSm - rainy summer
DA - dry autumn
RA - rainy autumn
DW - dry winter
FLW - winter floods
HSW - Heavy snowfall in winter
LMI - Locust/mouse invasions
EP - epidemics
IH - inflation and hunger
WH - weak harvests
AH - abundant harvests
ND - no data
CWe - cold weather
WWe - warm weather
AP - abundant precipitation
HS - heavy snow
SW - strong winds
SS - strong storms

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
