# Peer review of "Climatic extremes and their social impact in 17th-century Transylvania. A climate-historical reconstruction in the context of the Little Ice Age"

_EGUsphere, 2025_

## Referee Comment (RC2)

**Referee Report:**

**Title:** *Climatic extremes and their social impact in 17th-century Transylvania. A climatic historical reconstruction in the context of the Little Ice Age*

**Authors:** Gaceu et al.

Journal*: EGUsphere*

Please note that I deliberately did not read any of the possible postings of comments for this submission so as not to be influenced in my evaluation/opinion of this paper.

This paper presents a primarily documentary-based approach to examining the 17th century climate of Transylvania and compares these finding with some other proxy records for the region. The paper also places the findings within existing literature which cover the climate for this period in other parts of Europe in particular. I personally find this kind of work most interesting and valuable but also appreciate the enormous efforts (and time!) required to accumulate enough data (information) from which to present a meaningful contribution to the subject matter. This paper presents some valuable (new) information for a period and region, for which past weather and climate during the 17th century is still 'patchy' – to this end I would like to see some of this work published. However, there are several major concerns I have with the current manuscript, that would not make it suitable for publication in its current form. My aim here is to guide the authors toward revising their manuscript so that it would indeed make it a valuable publication for *EGUsphere*.

1. The manuscript is exceptionally long, written in a longwinded language style, and is also repetitive in style. For example, the lengthy *conclusion* (almost a full page of text) reads not as a conclusion but as a summary and merely repeats (summarizes) what had already been said in the paper. While a summary focuses on recapping information (which is what we have for this paper), a conclusion emphasizes the implications and deeper thoughts derived from the information presented (the current paper does not achieve this). Another illustration is perhaps with the title of the manuscript, which too is longwinded in style. It would read more concisely as: *Climatic extremes and their social implications in 17th-century Transylvania: An historical climate reconstruction*. There are many parts of the paper that could be cut out or substantially trimmed down.

2. I appreciate that English would not be the authors' primary language. However, the many typological/grammatical errors throughout the manuscript and the

longwinded language style as mentioned above, means that very heavy language editing is required and that the paper needs to be substantially trimmed down in length. There is also inconsistency with issues such as the use of upper- and lower-case lettering (for example western and Western Europe etc).

**3. Methodology:**

a) I note the rather large data gaps for especially climate-related risk phenomena & temperature conditions, and to a lesser extent rainfall. I am concerned that the results presented have not fully taken these very large data gaps into consideration and the results are undoubtedly impacted by these. The data gaps may not only be due to an absence of extreme cold, heat, drought or flooding, but could of course also be a consequence of other societal-based reasons (e.g. times of war, conflict etc). I would deem it essential to provide a graph illustrating the annual number of sources from which weather/climate information was obtained, and then to more critically address the data quality/quality & gaps issues in a temporal context, and how this might impact on the results.

b) I appreciate the efforts taken to present the temporal network information presented in Figures 3, 5 & 8. However, these Figures (especially Figure 5 & 8) are very 'busy' and not at all easy to appreciate in terms of extracting something that is important and meaningful for what the paper aims to achieve. Neither do the authors engage much with these 'temporal networks' in the written text and demonstrate their importance for paper. I strongly advise these be removed from the paper.

c) Figure 7 can also be cut out as again it is not easy to appreciate. I suggest that the cold and heat waves section be trimmed down and creatively incorporated into the section on 'cold and warm years and seasons' – this would help avoid some element of repetition and help make it less 'lengthy'.

d) Perhaps my biggest frustration with this paper is the extensive use of code language. Excluding the well-known acronyms such as LIA, NAO etc (which are widely known and most acceptable to use as acronyms), the reader must become familiar with no fewer than 36 codes (acronyms) specifically designed for this paper only (these are explained in the lengthy 3.2 section of the manuscript). From there, on one must then remember all these codes. This makes the reading task very heavy and tiresome. Such a code language style must be avoided.

4. General focus of discussion:

The paper places much of its focus on the fact that we are dealing with the Maunder Minimum (MM) and that broadly the results through the 17[th] century reflect the solar forcing (sunspot) influence – i.e. that with fewer sunspots as the century progresses there is a general cooling & associated changes with precipitation. The paper illustrates the agreement with other proxies and findings from other parts of Europe. So, the investigation takes a rather broad (general)

approach and essentially does not make any novel new findings that might expand knowledge for the 17[th] century northern hemisphere (Europe). There is much more to it than merely solar forcing, as many papers have demonstrated for 17[th] century Europe. The inability for the current paper to address smaller temporal scale (inter-annual) variability through their time of investigation, is a major limitation. For example, volcanic forcing in both the first 50 years and second 50 years of the 17[th] century is well known through several publications, yet the current paper makes no mention of such. I would deem it essential to more carefully consider volcanic forcing, especially given the fact that this paper is most concerned with 'climatic extremes', which we know often follow the shorter-term (~1-4 years) climatic impacts of volcanic eruptions, rather than the longer term (decades) effects of solar forcing changes. What about other ocean-atmospheric interactions that may have caused some of the observed temperature and hydro extremes? It is also well documented that the MM did indeed experience colder than normal winters but also at times rather unusually warm summers in Europe– this is not something new and so the authors really need a stronger connect with the literature that has addressed some of these things.

**A few smaller technical matters:**
Avoid vagueness: Line 23: 'Correlating historical sources' = very vague – what are these sources? I assume you mean 'Correlating documentary sources with other proxy data....'.

You refer to 'social archive' and 'society's archives' in some places....I think what you mean to say is 'archives of society' as the former two terms do not make sense.

You refer to 'altitude' – this is not correct when dealing with land above a certain height above sea level – for which it is not 'altitude' but 'elevation'. Altitude refers to the height in the atmosphere above the land surface (for example the flight altitude of an airplane).

'What are 'fohn movements'? = vague. Is this the same as what is more commonly referred to as 'föhn' – which is a warm wind?

I note the excessive use of the term 'database' – in most such cases it should be 'data'. A database is an organized collection of data that are filed nowadays in an electronic system of sorts. There seems a lot of confusion with such terms – for example you refer to disadvantages of 'databases' in line 158 – in this case what you are really dealing with is the disadvantage of 'data source types' and not

databases. You also discuss what you refer to as 'true databases' and 'primary databases' etc, which again are all terminologically incorrect for the context of discussion. This terminological confusion is widespread through the manuscript.

Please double check spelling – I note at least one error in Figure 2 ('regim' should be 'regime').

Lines 190 to 195 you write about the shortcoming of the 'methodology' when in fact the shortcoming has to do with the 'source type', rather than the methods. If you know that there are limitations with the source type, then you can explain how the methods have creatively dealt with such a limitation.

As mentioned already, I do not like the code language used for reasons already explained. But apart from that, some of the allocated acronyms (codes) do also not make much sense as there is inconsistency in the lettering allocation system you use. For example: for **c**old **y**ears you use (CY), yet for **w**arm **y**ears you use (HY)...why is it not WY? Yet in other places you then also use the code HY for **h**ot **y**ear. So, this means you have both warm years and hot years ...yet I do not see these distinctions in the results as there you only have hot years with warm seasons. Also, how do you differentiate between a hot year and a warm year? What qualifies a year as 'hot'? Do warm multiple seasons in a given year then make that year a hot year?...or should it not rather simply be a warm year? I pose all these questions because I can see there is a lot of confusion and mix-up in the paper with regards all this.

Figure 4: there is again some terminological confusion or mix-up here. I have always understood season to be summer, autumn, winter, spring. Here the Figure separates two of the seasons i.e. a = 'summer season' and b = 'winter season'.  You then use the dark colour to indicate 'cold season' (i.e. winter) and 'hot season' (i.e. summer). So, I think you are really referring to the **mean condition** of a given season, so for the colour boxes it should read as 'colder than normal conditions' and 'warmer than normal conditions'.

Line 337: 'The trend of winters becoming colder and more frequent was.....'. It is impossible for there to be a trend in winter frequency. There is always only one winter season per annum and so what you say here is technically impossible. There may indeed be trends for the thermal conditions of winters, or also the length of abnormally cold conditions making the winters *feel* either shorter or longer each year.

I hope my comments are constructive and will assist the authors to substantially improve their manuscript for possible publication in *EGUsphere*. I wish the authors well as they revise their work.

---

## Referee Comment (RC3)

**Review of the *Climate of the Past* manuscript:**

**Climatic extremes and their social impact in 17th-century Transylvania. A climate-historical reconstruction in the context of the Little Ice Age**

by Ovidiu Razvan Gaceu, Tudor Caciora*, Mihai Dudaș, Ștefan Baias, Marius Stupariu, Maria Maxim, Cătălina Mărculeț

**General comments**:

I appreciate the efforts of the authors to compile this impressive historical data base. However, there remains a lot to do concerning the presentation of the results and – in particular – the language of the paper.

(1) Many scientific terms used in this paper are either unusual – or not fully correct. Below you find some examples, but this list is not exhaustive.

Line 14: "thermal values" – why not "temperatures" ? ("thermal values" are used to quantify insulating properties)

Line 47: "individualized" is generally understood as a synonym for "personalized" – that's not what you wanted to say about the MM.

Line 91: What are "centralised studies" ?

Line 94: "when meteorological observations were not available" I would suggest to use "measurements" instead of "observations" – since observations (described in historical archives) are in fact the basis of your analysis.

Line 97: "experiential dimension of climate". I would understand "experiential" as "based on experience", like in "experiential learning".

Figure 1, caption: "territorial expansion". "Expansion" means an **increase** in size – that's not what you want to say.

Figure 1: "Altimetry" is the science of the **science** of measuring altitudes/elevations …
I would suggest to use "elevation" instead.

Line 131: I am aware that "Pontic" refers to the "Black Sea", however, I have never seen "Pontic air mass" mentioned in scientific literature.

Line 136: "fohn movements" I guess that you mean the "Föhn effect"

Line 257: "glacier in the Scarisoara cave". The ice in this cave is certainly not a glacier. Glaciers are formed by compression of snow – and they move (by definition).

Figure 10: "solar points" should be "sunspots" (and not "sun spots").

(2) The presentation of the results needs to be improved. Several examples are included in the (again non exhaustive) list of specific comments below. I understand that acronyms are necessary, but the inconsistent use (see specific comments) makes it particularly hard to follow the argumentation. Another example: In Fig. 4 "violet" means "cold" and yellow"

means "warm" (which is fine). However, in Fig. 3 immediately above "yellow" means "cold" **and** "warm". Similar in Fig. 5: Please use different colors for cold, warm, dry and excess (precipitation). In Fig. 6 "no data" s represented by "green" in (b) and by "yellow" in (a) – please avoid! Figure 7 is particularly hard to read.

(3) The attribution of the climate variations to solar forcing is too exaggerated (e.g., "The most plausible explanation for the high frequency of very low temperatures during the 17th century is low solar activity.", line 594) and not backed by the results shown. This attribution is also not really supported by the cited references. E.g., Stangl and Foelsche (2022) conclude: "This comparison suggests a certain solar influence but the agreement is not very pronounced." And they found "an unusually **small** number of severe winters during the last decades of the MM" (for the same study area) – which does not speak for a strong solar influence.

(4) Please provide references for all the software packages used.

(5) References: Please provide DOIs for all the references.

**Specific comments**:

Line 20: "of which 36 occurred during the Maunder Minimum (1645-1715)." Do you actually mean in the period 1645-1715, or in the period 1645-1700 (which would be the end of your study period)?

Line 23: "from the natural archive" --> "from natural archives" (there is not just one).

Line 40: "In this regard, Perșoiu et al. (2017) mention that in the first part of the Holocene …" Is the first part of the Holocene really relevant for your study period?

Line 48: "2°C" --> "2 °C" – you should always use space between number and unit.

Line 60: "cores" – do you mean "ice cores" ?

Line 83: "The hypothesis from which it was started is that, in agreement with western and central Europe, the territory of Transylvania must have been characterized by a similar climate …" Is this really true? It is contradicted by the statements in the following lines. And isn't one of the values of your study, that you can characterize differences to the climate in Western and Central Europe?

Figure 1: The color bar for the elevation ("altimetry") is a bit misleading, sin the colos don't match those of the map. Furthermore, it starts with "-20 m". I am aware that parts of the Danube Delta are below seal level – but the map doesn't show this part of modern Romania.

Figure 1: "Localities referred to in the"  There is something missing.

Methods: Apparently, the method applied is (understandably) very similar to the one used by the same authors (Gaceu et al, 2025) for the 16[th] century. I wonder why this is not mentioned in the text.

Line 177: "literate people, with writing concerns." I would understand "Writing concerns" as "Writing difficulties". I don't think that you wanted to say this.

Figure 2: "Climat" --> "Climate"; "regim" --> "regime"

Line 215: "cold (CY) and warm (HY) years". According to Fig. 2, HY should stand for "hot" year (not "warm").

Line 215: "cold (HW) … winters". According to Fig. 2 "H" stands for "harsh" – and this not a very good choice, since "H" can also mean "hot" sea above.

Line 220: "heavy precipitation (AP)". According to Fig. 2 this would by "abundant" precipitation. If it should stand for extreme precipitation than I would suggest to use " heavy precipitation" with the acronym "AP".

Line 225: "high prices and famine (IH)". According to Fig. 2 "IH" stands for "inflation and hunger"

Line 257: $\delta^{18}$O have been used – as a proxy for what?

Line 276: $\delta^{13}$C have been used – as a proxy for what?

Figure 4: "no data". This is a bit misleading. As you described before (and after), this can also mean that nothing was mentioned, because the weather was "normal". If there should be real data gaps, I would suggest to mark them in a different color.

Figure 4: How is, e.g., the "Winter 1601" to be understood. Is this the winter 1600/1601 or the winter 1601/1602 ?

Line 321: In the second half of the century, when the cooling was more pronounced, winters were reported as being extremely cold, 36 out of the 55 winters ..". This is a bit misleading- The second **half** of the century should not include more than 50 winters, right?

Line 326: "the winter of 1645". Is this 1645/46 ?

Line 359: "In this century, 64 years, testimonies of these types of climatic phenomena were recorded." What does this mean?

Line 366: ".. anticyclones, most often united in winter," What does "United" mean in this context? Do you mean that different anticyclones merged?

Figure 9, caption: "b – famine and famine" ?

Figure 10: I cannot distinguish the colors for "warm spring" and "warm summer".
The figure shows more a "comparison" than a "correlation". If you would indeed compute a correlation, it would likely be small (e.g. 3 cold winters in the decade with lowest solar activity, but 4 in the decade with highest solar activity ?).

Figure 11: The description of the comparison with proxy data is not very clear. If I understand it correctly, low $\delta^{18}$O from Scărișoara Cave shall reflect cold winters (line 621: " a sharp decrease in winter temperature starting ~800 years ago, with a minimum during the MM and LIA, when $\delta^{18}$O values reached their lowest levels"). However, according to Fig. 11 the lowest $\delta^{18}$O values within the 17[th] century occurred **after** the MM – when the winters where **not** particularly cold. The absolute minimum occurs in 1619 – together with a warm winter.

Figure 12: Als the discussion related to this Fig. is not clear enough. First you report on $\delta^{18}$O values from stalagmites – but you don't show the data. Nevertheless, you find that "The data from these natural reconstructions fit perfectly with the data obtained in this study from historical documents", line 661). Then you jump to $\delta^{13}$C data – without explaining what they are supposed to tell us – which are actually shown in Fig. 12. If the "perfect fit" refers to this

Fig. – I cannot see it. And how could there be a perfect fit, when the $\delta^{13}C$ data change only gradually, while the historic data show much higher variability.

Line 734: I wonder if this "data availability statement" will be satisfactory for *Climate of the Past*.

Please note that this are just some examples. Any careful copy-editor will find more.

After all this criticism I want – nevertheless – to conclude that I regards this as an important study, which deserves to be published, if it is properly revised.

---

## Author Comment (AC5)

**Dear reviewer,**

**thank you for your comments and suggestions, which were very useful and helped us to draft a clearer, more scientifically sound and better graphically structured manuscript. We will do our best to implement all the proposed recommendations.**

1. The manuscript is exceptionally long, written in a longwinded language style, and is also repetitive in style. For example, the lengthy *conclusion* (almost a full page of text) reads not as a conclusion but as a summary and merely repeats (summarizes) what had already been said in the paper. While a summary focuses on recapping information (which is what we have for this paper), a conclusion emphasizes the implications and deeper thoughts derived from the information presented (the current paper does not achieve this). Another illustration is perhaps with the title of the manuscript, which too is longwinded in style. It would read more concisely as: *Climatic extremes and their social implications in 17th-century Transylvania: An historical climate reconstruction*. There are many parts of the paper that could be cut out or substantially trimmed down.

We acknowledge the reviewer's observation regarding the length of the manuscript, the sometimes redundant style, and the overly long title and conclusions section. We agree that a more concise and focused writing can improve the clarity and scientific impact of the paper and we are prepared to do so, but these changes should not compromise the meaning of the manuscript.

2. I appreciate that English would not be the authors' primary language. However, the many typological/grammatical errors throughout the manuscript and the longwinded language style as mentioned above, means that very heavy language editing is required and that the paper needs to be substantially trimmed down in length. There is also inconsistency with issues such as the use of upper- and lower-case lettering (for example western and Western Europe etc).

To address this issues, the manuscript will be fully reviewed by a native English speaker with experience in academic writing before the revised version is resubmitted.

3. Methodology:

    a) I note the rather large data gaps for especially climate-related risk phenomena & temperature conditions, and to a lesser extent rainfall. I am concerned that the results presented have not fully taken these very large data gaps into consideration and the results are undoubtedly impacted by these. The data gaps may not only be due to an absence of extreme cold, heat, drought or flooding, but could of course also be a consequence of other societal-based reasons (e.g. times of war, conflict etc). I would deem it essential to provide a graph illustrating the annual number of sources from which weather/climate information was obtained, and then to more critically address the data quality/quality & gaps issues in a temporal context, and how this might impact on the results.

We fully acknowledge the presence of data gaps in our historical dataset, particularly regarding precipitation, temperature, and climate-related risk phenomena. However, it is important to note that the availability of historical climate information in the territory of present-day Romania is extremely limited compared to Western Europe, for example.

Unlike regions such as France, where detailed annual records (e.g., grape harvest dates and others) allow precise climatic reconstructions, historical documentation in Transylvania and the Romanian Principalities is scarce. Here, most of the population during the 17th century was illiterate, and even among the literate Romanian minority, writing and record-keeping were not common practices. As a result, most of the information we have comes from foreign observers, whose focus was not always on environmental conditions (foreign travellers, people documenting battles, etc.).

We agree that the lack of recorded climate phenomena does not necessarily imply their absence, and that data availability may be influenced by broader societal contexts such as war, instability, or cultural factors. In accordance with the reviewer's valuable suggestion, we will provide a more in-depth discussion regarding the temporal distribution, quality, and limitations of the data, and how these issues might influence the interpretation of results.

b) I appreciate the efforts taken to present the temporal network information presented in Figures 3, 5 & 8. However, these Figures (especially Figure 5 & 8) are very 'busy' and not at all easy to appreciate in terms of extracting something that is important and meaningful for what the paper aims to achieve. Neither do the authors engage much with these 'temporal networks' in the written text and demonstrate their importance for paper. I strongly advise these be removed from the paper.

We understand the concern that especially Figures 5 and 8 may appear visually overwhelming and difficult to interpret at first glance. However, we respectfully consider that these figures are essential for the core analytical structure of our study. They are not simply illustrative, but fundamental to understanding the complex correlations between climatic indicators and their associated social impacts, which form the backbone of our historical climate reconstruction.

The purpose of these temporal network visualizations is to allow a synthetic interpretation of nearly one thousand historical testimonies, to identify how one variable influences another in a given temporal context. For example, outbreaks of plague are frequently correlated with periods of cold and wet weather, suggesting that these harsh climatic conditions may have contributed, directly or indirectly, to the deterioration of public health. Similarly, numerous episodes of excess precipitation are temporally associated with cold winters, indicating that during the Little Ice Age and especially the Maunder Minimum, winters in this region of Europe were not only colder but also wetter. Furthermore, episodes of food shortages and famines occur predominantly in years or seasons marked by severe cold...these examples can go on.

These diagrams serve as indispensable tools for detecting and visualizing such relationships in a systematic manner, which would be virtually impossible to present using only narrative text or traditional tables. Without them, it would be extremely difficult to interpret the dataset holistically or to derive meaningful conclusions about climate evolution and its socio-economic consequences in 17th-century Transylvania.

That being said, we acknowledge that these figures require a more accessible and focused explanation in the manuscript, but we cannot give up on them altogether. In response to the reviewer's concern, we will revise the manuscript to provide clearer, more coherent textual support that highlights the most relevant insights from each figure and guides the reader in interpreting the data. We are confident that, through these improvements, the role and value of the temporal network figures will become more evident and better aligned with the overall aims of the paper.

c) Figure 7 can also be cut out as again it is not easy to appreciate. I suggest that the cold and heat waves section be trimmed down and creatively incorporated into the section on 'cold and warm years and seasons' – this would help avoid some element of repetition and help make it less 'lengthy'.

We respectfully disagree with the suggestion to remove Figure 7. While we understand the reviewer's concern regarding visual complexity and textual repetition, Figure 7 plays a critical role in presenting the data in a comprehensive, synthetic, and accessible format.

This figure summarizes a total of approximately 400 distinct climate-related events, including 116 cold and heat wave episodes, 170 events linked to excess and deficit precipitation, and 102 climate risk phenomena. These events vary significantly in duration, from just a few days to several weeks or even an entire month. Integrating this volume of detailed information purely into the main text would result in a manuscript that is not only excessively long, but also difficult to follow and overwhelming for the reader. On the other hand, eliminating these events altogether would significantly weaken the scientific value and integrity of the study, as it would mean presenting only a partial and fragmented view of the climate variability documented in 17th-century Transylvania. Our aim is to provide a full picture of the diversity, frequency, and distribution of short-term extreme events, which are essential for understanding the complexity of the historical climate.

Therefore, we consider Figure 7 essential for presenting this body of data in a manageable visual format. That said, we fully acknowledge the need for clearer guidance to help the reader interpret the figure. In the revised version of the manuscript, we will improve the accompanying text by more explicitly explaining how to read and use the figure, highlighting key findings, and connecting the visual data to specific narrative insights.

d) Perhaps my biggest frustration with this paper is the extensive use of code language. Excluding the well-known acronyms such as LIA, NAO etc (which are widely

known and most acceptable to use as acronyms), the reader must become familiar with no fewer than 36 codes (acronyms) specifically designed for this paper only (these are explained in the lengthy 3.2 section of the manuscript). From there, on one must then remember all these codes. This makes the reading task very heavy and tiresome. Such a code language style must be avoided.

We fully agree with the reviewer's concern regarding the excessive use of internally defined abbreviations. To improve readability, we have removed all such abbreviations from the main text. They will be used only within figures, where necessary for space, and will be clearly explained in each figure legend. We believe this change will make the manuscript much easier to follow and appreciate the reviewer's helpful suggestion.

General focus of discussion:

4. The paper places much of its focus on the fact that we are dealing with the Maunder Minimum (MM) and that broadly the results through the 17th century reflect the solar forcing (sunspot) influence – i.e. that with fewer sunspots as the century progresses there is a general cooling & associated changes with precipitation. The paper illustrates the agreement with other proxies and findings from other parts of Europe. So, the investigation takes a rather broad (general) approach and essentially does not make any novel new findings that might expand knowledge for the 17$^{th}$ century northern hemisphere (Europe).

We respectfully argue that, while the paper may appear to take a broad approach, its main contribution lies in filling a significant regional knowledge gap. While the evolution of weather and climate during the 17th century is well-documented for much of Western and Central Europe through historical sources, very little is known about this period in the territories of the former Romanian principalities. This is largely due to the scarcity of studies and the fragmentary nature of available sources, which have previously made it difficult to construct a coherent scientific narrative without over-reliance on extrapolation.
Moreover, we emphasize in the manuscript that the Little Ice Age (LIA) and the Maunder Minimum (MM) did not manifest uniformly across the globe, or even across Europe (this is a widely recognized fact in the scientific world). Thus, our study contributes to understanding regional climatic variability during this period and offers a new dataset based on a wide range of documentary evidence that has not been systematically analyzed before for the area of Transilvania.
At the same time, although there are several proxy-based reconstructions targeting various climatic parameters for 17th-century Transylvania, historical documentary data provide an essential complementary perspective, as they not only reflect environmental variability, but also capture how such changes were perceived, interpreted, and experienced by contemporary societies. This human dimension, absent from most natural proxies, is crucial for understanding the broader societal impacts of climate fluctuations.

5. There is much more to it than merely solar forcing, as many papers have demonstrated for 17th century Europe. The inability for the current paper to address smaller temporal scale (inter-annual) variability through their time of investigation, is a major limitation. For

example, volcanic forcing in both the first 50 years and second 50 years of the 17th century is well known through several publications, yet the current paper makes no mention of such. I would deem it essential to more carefully consider volcanic forcing, especially given the fact that this paper is most concerned with 'climatic extremes', which we know often follow the shorter-term (~1-4 years) climatic impacts of volcanic eruptions, rather than the longer term (decades) effects of solar forcing changes.

Thank you for this important remark. We fully acknowledge that volcanic forcing had the potential to influence climate variability in 17th-century Europe, including the region of our study. We will revise the manuscript to explicitly mention this factor and its relevance in the broader paleoclimatic context.
However, we did not include a detailed analysis of volcanic impacts in our study for two main reasons:
(1) There is a lack of regional-scale quantitative data that would allow us to assess the magnitude and specific influence of individual volcanic eruptions on the local climate of Transylvania.
(2) Even if such data were available, it would be methodologically difficult to establish direct causal links between individual eruptions and the observed climate-related events in our historical dataset, particularly given the complex atmospheric circulation patterns and delayed or regionally variable impacts that volcanic events can have.
Nonetheless, we agree that volcanic forcing should be recognized as an important short-term climate driver, especially in relation to extreme events, and we will integrate this perspective into the revised discussion.

6. What about other ocean-atmospheric interactions that may have caused some of the observed temperature and hydro extremes?

We agree that large-scale ocean-atmospheric interactions play a key role in shaping regional climate variability. In the manuscript, we have included a dedicated discussion of the North Atlantic Oscillation (NAO) and its phases, emphasizing its potential influence on both temperature and precipitation extremes in Transylvania during the 17th century.

7. It is also well documented that the MM did indeed experience colder than normal winters but also at times rather unusually warm summers in Europe– this is not something new and so the authors really need a stronger connect with the literature that has addressed some of these things.

We agree that the alternation between colder winters and unusually warm summers during the Maunder Minimum is well documented at the European scale. However, the novelty of our study lies in the regional focus on Eastern Europe, specifically Transylvania, where the climatic expression of both the Little Ice Age and the Maunder Minimum remains far less understood.
Moreover, the study contributes original insights by exploring how people experienced and interpreted climate variability during this period, through a systematic analysis of historical documentary sources. This human dimension, combined with the underexplored geographical focus, provides new perspectives that complement existing literature.

**A few smaller technical matters:**

8. Avoid vagueness: Line 23: 'Correlating historical sources' = very vague – what are these sources? I assume you mean 'Correlating documentary sources with other proxy data….'.

   It means 'Correlating documentary sources with other proxy data….', indeed. We will correct the sentence.

9. You refer to 'social archive' and 'society's archives' in some places….I think what you mean to say is 'archives of society' as the former two terms do not make sense.

   Indeed. We will correct the expression.

10. You refer to 'altitude' – this is not correct when dealing with land above a certain height above sea level – for which it is not 'altitude' but 'elevation'. Altitude refers to the height in the atmosphere above the land surface (for example the flight altitude of an airplane). Indeed. We will correct the expression.

11. 'What are 'fohn movements'? = vague. Is this the same as what is more commonly referred to as 'föhn' – which is a warm wind?

   We want to refer to this wind, yes. We will correct the expression.

12. I note the excessive use of the term 'database' – in most such cases it should be 'data'. A database is an organized collection of data that are filed nowadays in an electronic system of sorts. There seems a lot of confusion with such terms – for example you refer to disadvantages of 'databases' in line 158 – in this case what you are really dealing with is the disadvantage of 'data source types' and not

   Indeed. We will correct the expression.

13. databases. You also discuss what you refer to as 'true databases' and 'primary databases' etc, which again are all terminologically incorrect for the context of discussion. This terminological confusion is widespread through the manuscript.

   Thank you for this important observation. We acknowledge the inappropriate and confusing use of terms such as "true databases" and "primary databases." We will carefully revise the manuscript to correct all incorrect terminology and ensure consistency and accuracy throughout the text.

14. Please double check spelling – I note at least one error in Figure 2 ('regim' should be 'regime').

   The correction will be implemented!

15. Lines 190 to 195 you write about the shortcoming of the 'methodology' when in fact the shortcoming has to do with the 'source type', rather than the methods. If you know that there are limitations with the source type, then you can explain how the methods have creatively dealt with such a limitation.

It will be implemented!

16. As mentioned already, I do not like the code language used for reasons already explained. But apart from that, some of the allocated acronyms (codes) do also not make much sense as there is inconsistency in the lettering allocation system you use. For example: for **c**old **y**ears you use (CY), yet for **w**arm years you use (HY)…why is it not WY? Yet in other places you then also use the code HY for **h**ot year. So, this means you have both warm years and hot years …yet I do not see these distinctions in the results as there you only have hot years with warm seasons. Also, how do you differentiate between a hot year and a warm year? What qualifies a year as 'hot'? Do warm multiple seasons in a given year then make that year a hot year?...or should it not rather simply be a warm year? I pose all these questions because I can see there is a lot of confusion and mix-up in the paper with regards all this.

We acknowledge these typo issues. However, since we have decided to remove all internally defined acronyms from the main text, we believe this will greatly improve the clarity and fluency of the manuscript. The terms will now be fully written out, and such confusion will be avoided.

17. Figure 4: there is again some terminological confusion or mix-up here. I have always understood season to be summer, autumn, winter, spring. Here the Figure separates two of the seasons i.e. a = 'summer season' and b = 'winter season'. You then use the dark colour to indicate 'cold season' (i.e. winter) and 'hot season' (i.e. summer). So, I think you are really referring to the **mean condition** of a given season, so for the colour boxes it should read as 'colder than normal conditions' and 'warmer than normal conditions'.

The figure refers to temperature anomalies relative to expected seasonal conditions. We will revise the figure caption and legend to clearly specify that the shading indicates "colder than normal conditions" and "warmer than normal conditions" for the respective seasons.

18. Line 337: 'The trend of winters becoming colder and more frequent was…..'. It is impossible for there to be a trend in winter frequency. There is always only one winter season per annum and so what you say here is technically impossible. There may indeed be trends for the thermal conditions of winters, or also the length of abnormally cold conditions making the winters *feel* either shorter or longer each year.

This is a mistake, which will be corrected in the revised version of the article.

**Thank you very much once again, and we are looking forward to your answer!**

**All the best!**

---

## Author Comment (AC6)

**Dear reviewer,**

**thank you for your comments and suggestions, which were very useful and helped us to draft a clearer, more scientifically sound and better graphically structured manuscript. We will do our best to implement all the proposed recommendations.**

Line 14: "thermal values" – why not "temperatures"? ("thermal values" are used to quantify insulating properties)

We agree with your suggestion. We will modify it accordingly.

Line 47: "individualized" is generally understood as a synonym for "personalized" – that's not what you wanted to say about the MM.

We agree with your suggestion. We will modify it accordingly.

Line 91: What are "centralised studies"?

We want to refer to the fact that existing studies do not consider the use of multidimensional data (e.g. temperature, precipitation, risk phenomena, etc.) over a whole century or over several centuries. There are a few studies that analyze at a local level or over a limited period of years, a certain climatic element. But the term can be replaced by simply "studies".

Line 94: "when meteorological observations were not available" I would suggest to use "measurements" instead of "observations" – since observations (described in historical archives) are in fact the basis of your analysis.

Implemented!

Line 97: "experiential dimension of climate". I would understand "experiential" as "based on experience", like in "experiential learning".

We agree with your suggestion. We will modify it accordingly.

Figure 1, caption: "territorial expansion". "Expansion" means an increase in size – that's not what you want to say.

We agree with your suggestion. We will modify it accordingly.

Figure 1: "Altimetry" is the science of the science of measuring altitudes/elevations …I would suggest to use "elevation" instead.

We agree with your suggestion. We will modify it accordingly.

Line 131: I am aware that "Pontic" refers to the "Black Sea", however, I have never seen "Pontic air mass" mentioned in scientific literature.

We agree that the term "Pontic air mass" is not standard in scientific literature. We will revise the text to use a more appropriate and widely recognized formulation, such as "Black Sea influence" or "air masses influenced by the Black Sea region", to avoid confusion.

Line 136: "fohn movements" I guess that you mean the "Föhn effect"

Yes. We will correct to be as direct as possible in expression.

Line 257: "glacier in the Scarisoara cave". The ice in this cave is certainly not a glacier. Glaciers are formed by compression of snow – and they move (by definition).

This is widely recognized in the specialized literature as the "glacier from the Scarisoara cave", although it is not a glacier in the classical geomorphological sense of the word (i.e. a moving mass of ice, flowing under its own weight, as in the case of glaciers in the Alps or Himalayas). Although it is a deprecated term, it is in fact a stationary accumulation of permanent underground ice, located in a cave; therefore, a block of perennial ice formed by the refreezing of water from infiltrations and snow, under favourable microclimate conditions. In the manuscript, we will refer to it as "perennial ice block" or "cave ice block", for better understanding.

Figure 10: "solar points" should be "sunspots" (and not "sun spots").

Of course. We will modify it accordingly.

(2) The presentation of the results needs to be improved. Several examples are included in the (again non exhaustive) list of specific comments below. I understand that acronyms are necessary, but the inconsistent use (see specific comments) makes it particularly hard to follow the argumentation. Another example: In Fig. 4 "violet" means "cold" and yellow"means "warm" (which is fine). However, in Fig. 3 immediately above "yellow" means "cold" and "warm". Similar in Fig. 5: Please use different colors for cold, warm, dry and excess (precipitation). In Fig. 6 "no data" s represented by "green" in (b) and by "yellow" in (a) – please avoid! Figure 7 is particularly hard to read.

We will revise the figures to present the available data in a more centralized and accessible manner, improving clarity and visual coherence. Regarding the use of acronyms, we have decided to remove them entirely from the main text. They will be retained only within the figures, where necessary for space constraints, and each acronym will be clearly explained in the respective figure legend.

While we acknowledge that Figure 7 may appear somewhat dense at first glance, we consider it to be highly illustrative and essential to the paper. It presents nearly 400 historical records, including 116 events related to cold and heat waves, 170 events associated with precipitation excess or deficit, and 102 climate-related risk phenomena. Despite its complexity, the figure provides a necessary overview of the temporal distribution and nature of extreme climatic events, which could not be effectively conveyed through text alone. We believe it is a key component in understanding the dataset and its implications.

(3) The attribution of the climate variations to solar forcing is too exaggerated (e.g., "The most plausible explanation for the high frequency of very low temperatures during the 17[th] century is low solar activity.", line 594) and not backed by the results shown. This attribution is also not really supported by the cited references. E.g., Stangl and Foelsche (2022) conclude: "This comparison suggests a certain solar influence but the agreement is not very pronounced." And they found "an unusually small number of severe winters during the last decades of the MM" (for the same study area) – which does not speak for a strong solar influence.

Regarding the reference to Stangl and Foelsche (2022), we note that their conclusion is based on a much more limited dataset compared to the one presented in our study. Our documentary database not only includes the sources they used, but also many additional and previously unutilized records, which allows for a broader and more detailed perspective. This expanded dataset likely explains the discrepancy in the reported frequency of severe winters.

That said, we maintain that the pronounced cooling observed during the Maunder Minimum is unlikely to be coincidental, especially given the consistency of the signal across multiple figures and datasets presented in our study. As emphasized in the text, the clustering of cold winters, increased reports of cold waves, and climate-related societal impacts during this period all support the idea of an underlying external forcing mechanism. While we recognize that other drivers (e.g., volcanic activity, ocean-atmosphere dynamics etc) may also have contributed, we consider that low solar activity remains a plausible and important contributing factor, particularly when viewed in the broader paleoclimatic context.

(4) Please provide references for all the software packages used.

We will do so.

(5) References: Please provide DOIs for all the references.

We will do so.

**Specific comments:**

Line 20: "of which 36 occurred during the Maunder Minimum (1645-1715)." Do you actually mean in the period 1645-1715, or in the period 1645-1700 (which would be the end of your study period)?

We mean between 1645 and 1700, which falls within our study period. We will point this out better in the text.

Line 23: "from the natural archive" --> "from natural archives" (there is not just one).

We agree with your suggestion. We will modify it accordingly.

Line 40: "In this regard, Perșoiu et al. (2017) mention that in the first part of the Holocene …" Is the first part of the Holocene really relevant for your study period?

We agree that the first part of the Holocene is not directly related to our study period. However, we included the reference to Perșoiu et al. (2017) to provide brief paleoclimatic context in the introduction. This helps situate the climatic dynamics of the 17th century within a broader temporal framework.

Line 48: "2°C" --> "2 °C" – you should always use space between number and unit.

We agree. We will modify it accordingly.

Line 60: "cores" – do you mean "ice cores"?

Yes. We will modify it accordingly.

Line 83: "The hypothesis from which it was started is that, in agreement with western and central Europe, the territory of Transylvania must have been characterized by a similar climate …" Is this really true? It is contradicted by the statements in the following lines. And isn't one of the values of your study, that you can characterize differences to the climate in Western and Central Europe?

You are correct in noting that the statement regarding similarity with Western and Central Europe may appear inconsistent with later discussions in the manuscript. In fact, as shown in our previous study (Gaceu et al., 2025), there is evidence that Transylvania exhibited a somewhat different climatic pattern during the 16th century, with numerous reports of warm weather even during periods when the Little Ice Age was strongly expressed in Western Europe. In the current study, we aimed to further explore whether this divergence persisted in the 17th century (probably we will do it in another article, considering that such an analysis would be very extensive and inappropriate to be introduced in the present manuscript, which is already quite long). However, since we do not perform a direct comparative analysis in this paper, we recognize that the initial phrasing may be misleading. We will revise the introductory statement to better reflect the actual scope and goals of the study.

Figure 1: The color bar for the elevation ("altimetry") is a bit misleading, sin the colos don't match those of the map. Furthermore, it starts with "-20 m". I am aware that parts of the Danube Delta are below seal level – but the map doesn't show this part of modern Romania.

The regions of Wallachia and Moldavia were intentionally masked with a neutral color to visually emphasize that, although historically part of the Romanian principalities, they are not included in the scope of this study. This is why the color bar and elevation scale may appear misleading. We will revise Figure 1 to more accurately reflect the geographical realities of the study area, adjusting both the color scale and the visual representation where necessary.
Also, it seems that the value of -20 m was an error generated by the use of the DEM, and this has been corrected.

Figure 1: "Localities referred to in the" There is something missing.

Yes, we will correct the sentence.

Methods: Apparently, the method applied is (understandably) very similar to the one used by the same authors (Gaceu et al, 2025) for the 16th century. I wonder why this is not mentioned in the text.

The methodology is similar. We will introduce the citation in the text explicitly, so as not to cause confusion.

Line 177: "literate people, with writing concerns." I would understand "Writing concerns" as "Writing difficulties". I don't think that you wanted to say this.

Yes. We will modify it accordingly.

Figure 2: "Climat" --> "Climate"; "regim" --> "regime"

We will modify it accordingly.

Line 215: "cold (CY) and warm (HY) years". According to Fig. 2, HY should stand for "hot" year (not "warm").

It was a typo. We will modify it accordingly.

Line 215: "cold (HW) … winters". According to Fig. 2 "H" stands for "harsh" – and this not a very good choice, since "H" can also mean "hot" sea above.

We agree with your suggestion. We will modify it accordingly.

Line 220: "heavy precipitation (AP)". According to Fig. 2 this would by "abundant" precipitation. If it should stand for extreme precipitation than I would suggest to use " heavy precipitation" with the acronym "AP".

We agree with your suggestion. We will modify it accordingly.

Line 225: "high prices and famine (IH)". According to Fig. 2 "IH" stands for "inflation and hunger"

We will modify it accordingly.

Line 257: $\delta^{18}O$ have been used – as a proxy for what?

For winter temperatures. We will explain this fact better in the main text.

Line 276: $\delta^{13}C$ have been used – as a proxy for what?

For precipitation. We will explain this fact better in the main text.

Figure 4: "no data". This is a bit misleading. As you described before (and after), this can also mean that nothing was mentioned, because the weather was "normal". If there should be real data gaps, I would suggest to mark them in a different color.

"No data" may include multiple possible situations. However, we cannot confidently distinguish between years with truly normal weather (and therefore no mention), and those where data is missing due to other factors, such as lost manuscripts, lack of interest in recording the weather, or unrelated historical disruptions. Because of this uncertainty, we chose to use a single "no data" category. That said, we will clarify this limitation better in the main text to avoid any potential misunderstanding.

Figure 4: How is, e.g., the "Winter 1601" to be understood. Is this the winter 1600/1601 or the winter 1601/1602?

Winter 1601 refers to the winter between 1601 and 1602. We will clarify it.

Line 321: In the second half of the century, when the cooling was more pronounced, winters were reported as being extremely cold, 36 out of the 55 winters ..". This is a bit misleading- The second **half** of the century should not include more than 50 winters, right?

You are correct, the second half of the 17th century formally includes only 50 winters. Our reference was based on the Maunder Minimum period, which is commonly considered to have started around 1645. Therefore, the 55 winters mentioned refer to the interval 1645–1700, which slightly exceeds the calendar definition of "second half of the century." We will clarify this distinction in the revised manuscript to avoid any confusion.

Line 326: "the winter of 1645". Is this 1645/46?

Yes, it is 1645/1646.

Line 359: "In this century, 64 years, testimonies of these types of climatic phenomena were recorded." What does this mean?

We wanted to say that, in this century (17[th]), a total of 64 years were identified as being marked by the occurrence of cold or heat wave events.

Line 366: ".. anticyclones, most often united in winter," What does "United" mean in this context? Do you mean that different anticyclones merged?

Yes. We will explain this fact better in the main text.

Figure 9, caption: "b – famine and famine" ?

It supposed to be inflation and hunger. We will correct.

Figure 10: I cannot distinguish the colors for "warm spring" and "warm summer". The figure shows more a "comparison" than a "correlation". If you would indeed compute a correlation, it

would likely be small (e.g. 3 cold winters in the decade with lowest solar activity, but 4 in the decade with highest solar activity ?).

We will adjust the colour to be easier to distinguish. Regarding the correlation, we highlight not only the clear increase in the frequency of cold winters, for instance, between 1655 and 1665, all ten winters were reported as particularly cold, but also the rising number of accounts describing cold waves and exceptionally cold conditions during other seasons, especially during the transitional periods (spring and autumn).

Figure 11: The description of the comparison with proxy data is not very clear. If I understand it correctly, low $\delta^{18}$O from Scărișoara Cave shall reflect cold winters (line 621: " a sharp decrease in winter temperature starting ~800 years ago, with a minimum during the MM and LIA, when $\delta^{18}$O values reached their lowest levels"). However, according to Fig. 11 the lowest $\delta^{18}$O values within the 17th century occurred **after** the MM – when the winters where **not** particularly cold. The absolute minimum occurs in 1619 – together with a warm winter.

As we also noted in the manuscript, the correlation between documentary evidence and the $\delta^{18}$O proxy record from Scărișoara Cave is partial but not complete. This lack of full alignment may be explained by several factors. First, the ice from Scărișoara represents a mountain cave environment, which may not always reflect the same thermal signals captured in historical accounts originating from lowland or plains regions, often several hundred km away.

Second, local microclimatic conditions or regional topographic influences may result in discrepancies between the isotopic signal and broader-scale weather reports. Furthermore, some studies suggest that cave ice (and not only) can exhibit a delayed isotopic response to external climate forcing, which might help explain cases such as the low $\delta^{18}$O value in 1619 coinciding with a reported warm winter.

Despite these limitations, the Scărișoara record remains the only available continuous winter temperature proxy for Transylvania, and while it may not perfectly match all historical testimonies, it still provides valuable large-scale climatic insight for the region. We will clarify these aspects further in the revised text.

Figure 12: Als the discussion related to this Fig. is not clear enough. First you report on $\delta^{18}$O values from stalagmites – but you don't show the data.

The $\delta^{18}$O values referenced in the text are derived from ice deposits in Scărișoara Cave, not from stalagmites. We did not include these values in the figure because they are already published and thoroughly discussed in the study by Perșoiu et al. (2017), which we cited in the manuscript. The complete dataset is publicly available in that publication and its associated supplementary materials.

Nevertheless, you find that "The data from these natural reconstructions fit perfectly with the data obtained in this study from historical documents", line 661). Then you jump to $\delta^{13}$C data – without

explaining what they are supposed to tell us – which are actually shown in Fig. 12. If the "perfect fit" refers to this Fig. – I cannot see it. And how could there be a perfect fit, when the $\delta^{13}C$ data change only gradually, while the historic data show much higher variability.

To clarify, Figure 12 presents data from the OWDA (Old World Drought Atlas), which are tree-ring based reconstructions of precipitation variability. The figure shows the maximum, mean, and minimum values of reconstructed precipitation for the Transylvanian region, as extracted from the OWDA dataset. We will also add to the supplement the raw data used.

The reference to $\delta^{13}C$ was not intended to suggest a perfect year-to-year match with historical data. Rather, we aimed to highlight a general trend, particularly the increase in precipitation excess during the Maunder Minimum, especially in its later phase, a pattern that is supported both by the OWDA data and by the historical records.

We agree that $\delta^{13}C$ values generally change more gradually due to the nature of the proxy and the calibration method used. In contrast, the OWDA precipitation data have annual resolution, which naturally results in more pronounced interannual variability.

To avoid further confusion, we will revise the discussion in the manuscript to better distinguish between the different datasets used and the nature of the correlations being presented.

Line 734: I wonder if this "data availability statement" will be satisfactory for *Climate of the Past*.

The statement is wrong, considering that we included all the data we used in the supplement, the database is open access, and we are open to making it public. We will change the statement in the revised version of the manuscript.

**Thank you very much once again, and we are looking forward to your answer!**

**All the best!**